# Preference Optimization with Multi-Sample Comparisons

## Abstract

Recent advancements in generative models, particularly large language models (LLMs) and diffusion models, have been driven by extensive pretraining on large datasets followed by post-training. However, current post-training methods such as reinforcement learning from human feedback (RLHF) and direct alignment from preference methods (DAP) primarily utilize single-sample comparisons. These approaches often fail to capture critical characteristics such as generative diversity and bias, which are more accurately assessed through multiple samples. To address these limitations, we introduce a novel approach that extends post-training to include multi-sample comparisons. To achieve this, we propose Multi-sample Direct Preference Optimization (mDPO) and Multi-sample Identity Preference Optimization (mIPO). These methods improve traditional DAP methods by focusing on group-wise characteristics. Empirically, we demonstrate that multi-sample comparison is more effective in optimizing collective characteristics (e.g., diversity and bias) for generative models than single-sample comparison. Additionally, our findings suggest that multi-sample comparisons provide a more robust optimization framework, particularly for dataset with label noise.

## 1 Introduction

Generative models, particularly large language models (LLMs) (Achiam et al., 2023; Meta AI, 2024; Bai et al., 2023; Bi et al., 2024; Gemini et al., 2023; Anthropic, 2024) and diffusion models (Sohl-Dickstein et al., 2015; Ho et al., 2020; Rombach et al., 2022; Podell et al., 2023), hold the tremendous promise to transform numerous industries by automating complex tasks, enhancing creativity, and personalizing user experiences at an unprecedented scale (Eloundou et al., 2023). These models achieve their capabilities through extensive pretraining on large-scale datasets, followed by post-training to unlock the capabilities (i.e., the superficial alignment hypothesis) (Wei et al., 2022; Tay et al., 2022; Zhou et al., 2024a). For post-training stage, reinforcement learning from human feedbacks (RLHF) and direct alignment from preference methods (DAP) have been crucial for the success of both LLMs (Ouyang et al., 2022; Rafailov et al., 2024; Azar et al., 2024) and diffusion models (Black et al., 2023; Wallace et al., 2023; Yang et al., 2024).

Despite these advancements, current post-training approaches predominantly focus on single-sample comparisons, which fail to capture characteristics better assessed through distributions of samples, such as creativity and bias. Evaluating a model's creativity/consistency or detecting biases requires analyzing the variability and diversity across multiple outputs, not just individual ones. For example, while LLMs are proficient in crafting narratives, they often show limitations in generating a diverse representation of genres (Patel et al., 2024; Wang et al., 2024). Additionally, these models tend to have lower entropy in their predictive distributions after post-training, leading to limited generative diversity (Mohammadi, 2024; Wang et al., 2023; Wiher et al., 2022; Khalifa et al., 2020).

Consider the task of generating a random integer between 0 and 10. Ideally, the model should not prefer any particular number. However, Zhang et al. (2024a) show that existing models often bias towards certain numbers over the others. Similar issues are observed in diffusion models, which may display biases in the generated outputs based on factors like gender or race (Luccioni et al., 2023; Chen et al., 2024). Inconsistencies in generation is also a crucial issue that needs to be addressed to make models more reliable (Liu et al., 2023; Bubeck et al., 2023). The aforementioned failures cannot be captured by a single sample; instead, they are distributional issues (see Fig. 1 for an illustration). To address the limitations, we extend the post-training paradigm to multi-sample

Figure 1: **Top**: Diversity of responses from two groups for improving urban transportation. The left group provides a broader range of approaches, including public transit and infrastructure improvements. The right group focuses more narrowly on specific technological and management solutions. **Bottom**: Bias in images from two groups. The left group displays a more balanced representation of race and gender, while the right group predominantly features males due to stereotypes.

comparisons. This approach involves evaluating the model's performance over distributions of samples rather than individual samples. By curating groups of responses and assessing their collective characteristics, we can better align the model's outputs with desired distributional properties.

In this work, we introduce Multi-sample Direct Preference Optimization (mDPO) and Multi-sample Identity Preference Optimization (mIPO), which are extensions of the prior DAP methods DPO (Rafailov et al., 2024) and IPO (Azar et al., 2024)[1]. Unlike their predecessors, which rely on single-sample comparisons, mDPO and mIPO utilize multi-sample comparisons to better capture group-wise or distributional characteristics. Our experiments involve fine-tuning language models to generate random numbers in a calibrated manner and enhancing the diversity of genres in creative story writing. Additionally, for diffusion models, we demonstrate a significant reduction in gender and race biases compared to the traditional single-sample comparison approach. Furthermore, we present evidence that multi-sample comparison offers a more robust optimization method in the presence of label noise in preference data, such as synthetic data that lacks human labeling but possesses knowledge of the overall quality between two models.

## 2 RELATED WORKS

**Reinforcement Learning from Human Feedback (RLHF)** has been pivotal in advancing the capabilities of large language models (Ouyang et al., 2022; Bai et al., 2022) and, more broadly, in agent alignment (Christiano et al., 2017; Leike et al., 2018). RLHF was initially introduced by Christiano et al. (2017) to tackle classic reinforcement learning tasks such as those found in MuJoCo. The work by Ouyang et al. (2022) marked the first significant application of RLHF in aligning language models to follow human instructions, thereby establishing a foundation for subsequent models like ChatGPT (Achiam et al., 2023). To address the complexities and instabilities associated with RL methods, Direct Alignment from Preference (DAP) methods (Rafailov et al., 2024; Azar et al., 2024; Dong et al., 2023; Yuan et al., 2023; Zhao et al., 2023) have been proposed. Notably, Rafailov et al. (2024) introduced Direct Preference Optimization (DPO), which reframes the RL problem into a supervised learning problem by deriving an analytical relationship between policy and reward. To counteract potential overoptimization issues in DPO, Azar et al. (2024) proposed Identity Preference Optimization (IPO), which employs an $\ell_2$ loss to regress the margin towards a predefined threshold. In the context of diffusion models, Black et al. (2023) introduced Denoising Diffusion Policy Optimization (DDPO) to optimize diffusion models using specific reward functions. Wallace et al. (2023) expanded the application of DPO from language domains to image domains. However, these methods predominantly focus on single-sample comparisons and may not adequately capture the collective characteristics of output distributions.

---

[1]Our framework can also be extended to other preference optimization algorithms.

**Distributional difference** (Zhong et al., 2022) describes the difference between two or more distributions, which is a generalization of single-sample comparison. While many characteristics can be judged from a single sample, some properties can only be deduced from multiple samples, such as diversity and bias (Santurkar et al., 2023; Chen et al., 2024; Zhang et al., 2024b; Zhou et al., 2024b; Mohammadi, 2024; Wang et al., 2023; Go et al., 2023). To measure the distributional difference, Zhong et al. (2022) proposed a hypothesis-verifier framework to summarize the difference between two corpora. Dunlap et al. (2024) further extended this framework to the image domain to capture the difference between two sets of images. More recently, Zhong et al. (2023) considered the problem of distributional difference in a goal-driven setting. On the other hand, Melnyk et al. (2024) addressed the problem of distributional alignment for language models using optimal transport, aiming to optimize the chosen responses across all prompts jointly. In contrast, our work addresses an orthogonal problem by focusing on optimizing the distributional preference under the same prompt.

## 3 PRELIMINARIES

**Supervised Finetuning.** After the language model has been pretrained on extensive datasets, the next step is typically supervised finetuning (SFT). This process aims to refine the model's predictions, preparing it for subsequent alignment stages. During SFT, the pretrained language model is finetuned using a supervised dataset $\mathcal{D} = \{(x, y)_i\}_{i=1}^N$, where $x$ represents the input and $y$ denotes the target. The objective of SFT is to minimize the negative log-likelihood,

$$\mathcal{L}_{\text{SFT}}(\pi_{\boldsymbol{\theta}}, \mathcal{D}) = \mathbb{E}_{(x,y) \sim \mathcal{D}}\left[-\log \pi_{\boldsymbol{\theta}}(y|x)\right].$$

**Direct Alignment from Preference.** For single-sample comparison setting, the dataset consists of $\{(x, y_w, y_l)_i\}_{i=1}^N$, where $x$ denotes the prompt or the input to the model, $y_w$ is the preferred response over $y_l$. Under the Bradley-Terry model, the likelihood of $y_w$ is preferred to $y_l$ is computed by

$$p(y_w \succcurlyeq y_l|x) = \sigma(r(y_w, x) - r(y_l, x)) = \frac{\exp(r(y_w, x))}{\exp(r(y_w, x)) + \exp(r(y_l, x))},$$

where $r(y, x)$ is the scalar reward of answering $y$ when given $x$. Direct preference optimization (DPO) (Rafailov et al., 2024) establishes an analytical form between the reward function and the policy or the language model, i.e., the language model represents an implicit reward. Specifically, under DPO, the implicit reward can be computed by

$$r_{\boldsymbol{\theta}}(y, x) = \beta \log \frac{\pi_{\boldsymbol{\theta}}(y|x)}{\pi_{\text{ref}}(y|x)} + \text{const}(x).$$

Using the implicit reward, DPO optimizes the policy $\pi_{\boldsymbol{\theta}}$ by minimizing the negative log-likelihood on the offline preference dataset,

$$\mathcal{L}_{\text{DPO}}(\pi_{\boldsymbol{\theta}}, \mathcal{D}) = \mathbb{E}_{(x,y_w,y_l) \sim \mathcal{D}}\left[-\log \sigma\left(r_{\boldsymbol{\theta}}(y_w, x) - r_{\boldsymbol{\theta}}(y_l, x)\right)\right].$$

Although the Bradley-Terry model is based on the assumption of Gumbel noise in the reward function, a choice that is well-suited for discrete random variables, it has certain limitations, such as overfitting, as noted by Azar et al. (2024). Specifically, Identity Preference Optimization (IPO) (Azar et al., 2024) addresses these limitations by bypassing the Bradley-Terry model for preference modeling. IPO aims to mitigate the issue of "overfitting" in preference datasets by replacing the sigmoid function with the squared distance,

$$\mathcal{L}_{\text{IPO}}(\pi_{\boldsymbol{\theta}}, \mathcal{D}) = \mathbb{E}_{(x,y_w,y_l) \sim \mathcal{D}}\left[\log \frac{\pi_{\boldsymbol{\theta}}(y_w|x)}{\pi_{\text{ref}}(y_l|x)} - \log \frac{\pi_{\boldsymbol{\theta}}(y_l|x)}{\pi_{\text{ref}}(y_l|x)} - \frac{\tau^{-1}}{2}\right]^2,$$

which can be interpreted as regressing the difference between the log-likelihood ratio to $\tau^{-1}/2$.

## 4 METHOD

The singleton preference (i.e., the marginal distribution) may not generalize well to the distributional preference (i.e., the joint distribution) (Melnyk et al., 2024). For most of the cases, when given a prompt $x$, there is indeed a preference between two responses $y_1$ and $y_2$. However, there are also cases where the preference cannot be assigned purely based on two singletons. Consider the cases where we prompt the diffusion models to generate an image of "a software engineer". For such cases, there is no preference between "a female software engineer" and "a male software engineer". Similarly, if a language model is prompted to generate an integer randomly and uniformly from $\{0, 1, ..., 9\}$, there is also no preference between generating any two specific numbers, e.g., 3 or 5. Nonetheless, preferences can exist at a **multi-sample or joint distribution level**.

### 4.1 MULTI-SAMPLE PREFERENCE OPTIMIZATION

Thus, we generalize the preference model from singleton comparison to multi-sample comparison, where the preference is assigned at the multi-sample level instead of on a singleton-level. Let $\mathcal{G}_w$ and $\mathcal{G}_l$ denote the multiple samples sampled from the model's conditional distribution[2] when given the prompt $x$, then the likelihood of $\mathcal{G}_w$ is preferred over $\mathcal{G}_l$ is defined as

$$p(\mathcal{G}_w \succcurlyeq \mathcal{G}_l|x) = \Phi(r(\mathcal{G}_w, x) - r(\mathcal{G}_l, x)),$$

where the function $\Phi$ follows the definition in the IPO paper (Azar et al., 2024), which can be e.g., the sigmoid function (recovers the Bradley-Terry model). Given the reward function $r(\cdot, \cdot)$, the goal of RLHF is to maximize the reward under reverse KL constraints. This can be captured by the following optimization objective,

$$\max_{\boldsymbol{\theta}} \mathbb{E}_{\mathcal{G} \sim \pi_{\boldsymbol{\theta}}|x, x \sim p_x} \left[ r(\mathcal{G}, x) - \beta \cdot \mathrm{KL}(\pi_{\boldsymbol{\theta}}(\cdot|x)||\pi_{\mathrm{ref}}(\cdot|x)) \right].$$

Generalizing the DPO's derivation, we obtain the following equation for the reward function,

$$r(\mathcal{G}, x) = \beta \log \frac{\pi_{\boldsymbol{\theta}}(\mathcal{G}|x)}{\pi_{\mathrm{ref}}(\mathcal{G}|x)} + \mathrm{const}(x).$$

where $\mathrm{const}(x)$ is some constant that only depends $x$, and $\beta$ is the coefficient of the KL regularization. For a given dataset, $\{(\mathcal{G}_w, \mathcal{G}_l, x)_i\}_{i=1}^N$, where $\mathcal{G} = \{y_j\}_{j=1}^k$ with $y_j \overset{\mathrm{i.i.d}}{\sim} \pi(\cdot|x)$, we can expand the likelihood by $\pi_\theta(\mathcal{G}|x) = \prod_{y \in \mathcal{G}} \pi_{\boldsymbol{\theta}}(y|x)$. To avoid the effect of the size of $\mathcal{G}$, we consider using the geometric mean instead, $\pi_{\boldsymbol{\theta}}(\mathcal{G}|x) = (\prod_{y \in \mathcal{G}} \pi_{\boldsymbol{\theta}}(y|x))^{1/|\mathcal{G}|}$. Thus the objective of multi-sample DPO becomes (where $\Phi(z) = \sigma(z)$),

$$\mathcal{L}_{\mathrm{mDPO}} = \mathbb{E}_{(x, \mathcal{G}_w, \mathcal{G}_l) \sim \mathcal{D}} \left[ -\log \sigma \left( \beta \mathbb{E}_{y_w \sim \mathcal{G}_w} \left[ \log \frac{\pi_{\boldsymbol{\theta}}(y_w|x)}{\pi_{\mathrm{ref}}(y_w|x)} \right] - \beta \mathbb{E}_{y_l \sim \mathcal{G}_l} \left[ \log \frac{\pi_{\boldsymbol{\theta}}(y_l|x)}{\pi_{\mathrm{ref}}(y_l|x)} \right] \right) \right].$$

Similarly, for the IPO variant (where $\Phi(z) = (z - \tau^{-1}/2)^2$), we have

$$\mathcal{L}_{\mathrm{mIPO}} = \mathbb{E}_{(x, \mathcal{G}_w, \mathcal{G}_l) \sim \mathcal{D}} \left[ \left( \mathbb{E}_{y_w \sim \mathcal{G}_w} \left[ \log \frac{\pi_{\boldsymbol{\theta}}(y_w|x)}{\pi_{\mathrm{ref}}(y_w|x)} \right] - \mathbb{E}_{y_l \sim \mathcal{G}_l} \left[ \log \frac{\pi_{\boldsymbol{\theta}}(y_l|x)}{\pi_{\mathrm{ref}}(y_l|x)} \right] - \frac{\tau^{-1}}{2} \right)^2 \right].$$

The main difference (highlighted above in red) from their original objectives is that the implicit reward for a single sample is replaced with the averaged implicit reward for a group of samples. We use the notation $y \sim \mathcal{G}$ to account for the general case where $\mathcal{G}$ represents a distribution rather than merely a finite collection of samples.

### 4.2 STOCHASTIC ESTIMATOR FOR EFFICIENT OPTIMIZATION

Our focus is on finetuning large generative models, and thus a scalable estimator for the gradient / objective is necessary to make the algorithm practically useful. Deriving a minibatch (potentially unbiased and low-variance) estimator for mDPO is challenging due to the non-linearity of the sigmoid function. However, for mIPO, this task is more tractable by

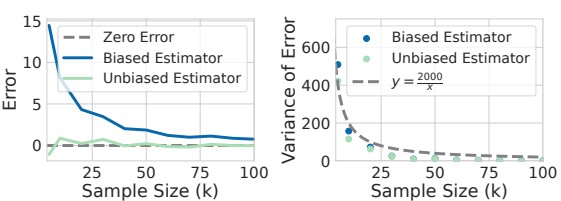

Figure 2: Biased estimator vs. Unbiased estimator.

expanding the square function. This is equivalent to computing the unbiased estimator for the following objective, $\ell = (\mathbb{E}_{x \sim p}[f(x)] - \mathbb{E}_{x \sim q}[f(x)] - c)^2$. The unbiased estimator for mIPO is stated in the following result.

**Proposition 1.** *Let $f : \mathcal{X} \to \mathbb{R}$ be a measurable function, and let $p$ and $q$ be probability distributions on $\mathcal{X}$. Define $\ell = (\mathbb{E}_{x \sim p}[f(x)] - \mathbb{E}_{x \sim q}[f(x)] - c)^2$, where $c$ is a constant. Let $x_1^p, \ldots, x_n^p$ be i.i.d. samples from $p$, and $x_1^q, \ldots, x_m^q$ be i.i.d. samples from $q$. Then,*

$$\hat{\ell} = \left( \frac{1}{n} \sum_{i=1}^n f(x_i^p) - \frac{1}{m} \sum_{j=1}^m f(x_j^q) - c \right)^2 - \left( \frac{\hat{\sigma}_p^2}{n} + \frac{\hat{\sigma}_q^2}{m} \right)$$

*is an unbiased estimator of $\ell$, where $\hat{\sigma}_p^2$ and $\hat{\sigma}_q^2$ are the sample variances of $f(x)$ under $p$ and $q$.*

---

[2]The definition of a group $\mathcal{G}$ can be generalized to distributions. Thus, $\mathcal{G}$ is equivalent to the predictive distribution of the language model.

Therefore, to optimize the mIPO objective, we can sample mini-batches similar to IPO. Each mini-batch can consist of multiple responses from both $\mathcal{G}_w$ and $\mathcal{G}_l$. While increasing the number of samples from each group reduces the variance of the estimated gradient, it also raises computational and memory costs. In summary, the mIPO objective is thus

$$\mathbb{E}_{(x,\{y_{w,i}\}_{i=1}^k,\{y_{l,i}\}_{i=1}^k)\sim\mathcal{D}}\left[\left(\frac{1}{k}\sum_{i=1}^k\log\frac{\pi_{\boldsymbol{\theta}}(y_{w,i}|x)}{\pi_{\text{ref}}(y_{w,i}|x)}-\frac{1}{k}\sum_{j=1}^k\log\frac{\pi_{\boldsymbol{\theta}}(y_{l,j}|x)}{\pi_{\text{ref}}(y_{l,j}|x)}-\frac{\tau^{-1}}{2}\right)^2-\frac{\hat{\sigma}_w^2}{k}-\frac{\hat{\sigma}_l^2}{k}\right].$$

For mDPO, we consider the following objective for multi-sample comparison, which may exhibit low variance (for larger values of $k$) but is biased,

$$\mathbb{E}_{(x,\{y_{w,i}\}_{i=1}^k,\{y_{l,i}\}_{i=1}^k)\sim\mathcal{D}}\left[-\log\sigma\left(\frac{\beta}{k}\sum_{i=1}^k\log\frac{\pi_{\boldsymbol{\theta}}(y_{w,i}|x)}{\pi_{\text{ref}}(y_{w,i}|x)}-\frac{\beta}{k}\sum_{j=1}^k\log\frac{\pi_{\boldsymbol{\theta}}(y_{l,j}|x)}{\pi_{\text{ref}}(y_{l,j}|x)}\right)\right].$$

To further understand the variance, we present the following Proposition 2, which shows that the variance of the estimator $\hat{\ell}$ is $\tilde{\mathcal{O}}(1/k)$, where $k$ is the number of samples.

**Proposition 2.** *Let $\mu_p = \mathbb{E}_{x\sim p}[f(x)]$, $\mu_q = \mathbb{E}_{x\sim q}[f(x)]$, $\sigma_p^2 = Var_{x\sim p}[f(x)]$, $\sigma_q^2 = Var_{x\sim q}[f(x)]$ and $n$ and $m$ be the number of independent samples from distributions $p$ and $q$, respectively. Then, the variance of the mini-batch estimator $\hat{\ell}$ is given by*

$$Var(\hat{\ell}) = \mathcal{O}\left(\left(\frac{\sigma_p^2}{n}+\frac{\sigma_q^2}{m}\right)\cdot\left(\frac{\sigma_p^2}{n}+\frac{\sigma_q^2}{m}+(\mu_p-\mu_q-c)^2\right)\right).$$

To gain an understanding about the variance and bias, we run a simulation using the function $f(x) = x^2$ to assess the effects of variance and bias and their relationship to the sample size $k$. The trend is plotted in Fig. 2. We observe that the unbiased estimator results in lower error compared to the biased estimator at small sample sizes. However, as the sample size increases, the error of the biased estimator also decreases and performs similarly to the unbiased one. This implies that as the sample size increases, the choice of estimator may become less critical.

## 5 EXPERIMENTS

### 5.1 RANDOM NUMBER GENERATOR (RNG)

Despite existing LLMs being trained to follow instructions and perform well on many tasks, they perform surprisingly poorly when asked to generate random outputs (See Fig. 3). A canonical example is instructing LLMs to generate a random number within a given interval. Existing LLMs exhibit a strong bias towards certain numbers rather than producing purely random outputs. In such settings, single-sample comparisons may be ineffective for modeling the distributional properties of the model's outputs, necessitating the use of multi-sample comparisons.

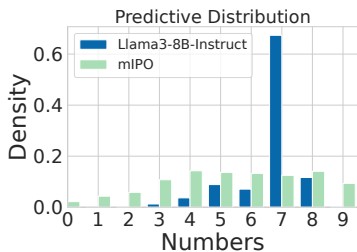

Figure 3: Dist. comparison.

**Preference data construction and finetuning.** Our initial experiments focus on generating integers uniformly at random within an interval $[a,b]$, where $a$ and $b$ are integers. To create the dataset, we sample $a$ randomly from $[0,1000]$ and the interval gap $m$ from $[5,10]$, with $b$ defined as $a+m$. The preferred response group is based on a uniform distribution of numbers, while non-uniform distributions are classified as rejected. We generated approximately 3,000 paired groups for training. For testing, we sample $a$ and $b$ from $[0,1000]$ with $a<b$ but do not impose the same gap constraints as in the training set to evaluate generalizability. We implemented the multi-sample versions of IPO and DPO, named mIPO and mDPO, respectively. To stabilize training, we add a negative log-likelihood[3] to the objective of both the multi-sample and original versions of IPO and DPO, shown to be effective in reasoning tasks (Pang et al., 2024). For finetuning, we use the Llama 3-8B instruct model with LoRA, rank 64, and $\alpha = 128$. More training details are in Appendix A.1.

---

[3]Without this, the language model tends to generate numbers beyond the specified range. We discuss this further from a constrained optimization perspective.

Table 1: Comparison of Win Rates in the Random Number Generation Experiment with Llama-3-8B: Win rates are calculated based on the uniformity of the distribution given the prompt.

| | mIPO vs IPO | mIPO vs SFT | IPO vs SFT | mDPO vs DPO | mDPO vs SFT | DPO vs SFT |
|---|---|---|---|---|---|---|
| Win Rate | 0.95 | 0.99 | 0.98 | 0.80 | 0.99 | 0.99 |

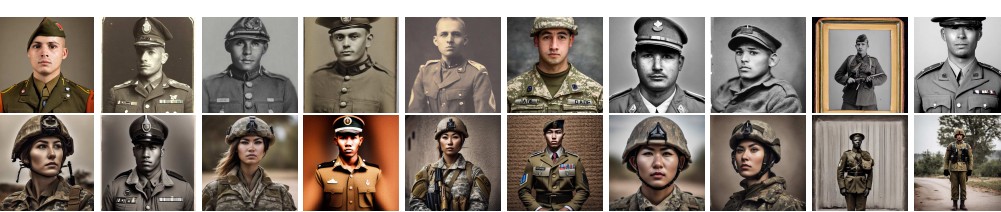

Figure 4: Qualitative comparison between images generated with the prompt *A portrait photo of a soldier*. The top row uses the original Stable Diffusion model, while the bottom row uses the mDPO-finetuned model, showing a more balanced distribution of race and gender.

**Metrics and results.** We compare the predictive entropy for each model with 100 testing prompts instructing the model to generate random numbers within specific intervals. We calculate the average win rates for the same prompt, with the winner being the one with larger entropy. The results are presented in Table 1. We observe that mIPO, IPO, mDPO, and DPO consistently and significantly outperform the SFT baseline. Additionally, both mIPO and mDPO outperform IPO and DPO by a large margin. Figure 3 shows the predictive distribution before and after finetuning. The original Llama3 model significantly favors the number 7 over others. However, after finetuning, the predictive distribution is much closer to the uniform distribution.

## 5.2 CONTROLLED DEBIASING FOR IMAGE GENERATION

Diffusion models (Sohl-Dickstein et al., 2015) have emerged as a powerful workhorse for image generation (Ho et al., 2020; Rombach et al., 2022; Dai et al., 2023; Esser et al., 2024). Despite their success, the generated images often reflect strong biases and stereotypes due to imbalances in the training data (Ananya, 2024). These biases can result in harmful societal stereotypes and limit the diversity of generated content. To illustrate this point, we visualize the distribution of generated images for different occupations and highlight the discrimination in race and biases in Figure 11 in the Appendix using Stable diffusion 1.5 (Rombach et al., 2022).

To address the generation biases, we extend the diffusion DPO objective proposed in Wallace et al. (2023) from single-sample comparison to multi-sample comparison in a similar way as we discussed in Section 4. For diffusion models, the mDPO objective can be formulated as[4]

$$\mathcal{L}_{\mathrm{mDPO}}^{\mathrm{diff}}(\epsilon_{\boldsymbol{\theta}}, \mathcal{D}) = \mathbb{E}\left[-\log \sigma\left(-\beta T \omega(\lambda_t) \cdot \left(\mathbb{E}_{\mathcal{G}_w}\left[r(\epsilon^w, \boldsymbol{x}_t^w, t; \boldsymbol{\theta})\right] - \mathbb{E}_{\mathcal{G}_l}\left[r(\epsilon^l, \boldsymbol{x}_t^l, t; \boldsymbol{\theta})\right]\right)\right)\right],$$

where $r(\epsilon^w, \boldsymbol{x}_t^w, t; \boldsymbol{\theta}) = \|\epsilon^w - \epsilon_{\boldsymbol{\theta}}(\boldsymbol{x}_t^w, t; c)\|_2^2 - \|\epsilon^w - \epsilon_{\mathrm{ref}}(\boldsymbol{x}_t^w, t; c)\|_2^2$, $\epsilon^w = \boldsymbol{x}_t^w - \boldsymbol{x}^w$ and $\boldsymbol{x}^w \sim \mathcal{G}_w$, and the expectation is taken over $(\mathcal{G}_w, \mathcal{G}_l, c) \sim \mathcal{D}$ and $t \sim \mathcal{U}(0, T)$. The same applies to $\mathcal{G}_l$, $\boldsymbol{x}_t^l$ and $\boldsymbol{x}^l$. We adopt the Stable Diffusion 1.5 (Rombach et al., 2022) as our base model due to its popularity within the community, which we also find has strong bias in favor of certain genders and races.

**Dataset construction and finetuning.** To construct the dataset, we first collect 50 occupations, and take 80% of the occupations for generating training data and the remaining 20% for evaluation. For each occupation, we generate 6 images with varied races in {black, white, aisan} and genders in {male, female}, which form our chosen group. For the rejected group, we use the images generated from the original Stable diffusion 1.5 via API[5]. Our training code is adapted from the original implementation by Wallace et al. (2023). More details on the training can be found in Appendix A.2.

**Evaluation metric and results.** To measure the generation quality, we used the Simpson Diversity Index (Simpson, 1949), which focuses more on the dominance and even distribution of species. To compute it, we use $D = 1 - \sum_i (n_i/N)^2$, where $n_i$ is the number of instance falls in $i_{\mathrm{th}}$ category (e.g., for gender, the categories we consider are {male, female}), and $N = \sum_i n_i$. In

---

[4]It's more appropriate to interpret $r(\epsilon, \boldsymbol{x}; \theta)$ as a cost rather than a reward. Consequently, the formulation uses $-\beta$ instead of $\beta$, as done in the original DPO.

[5]https://stablediffusionapi.com/docs/category/stable-diffusion-api

general, the higher the value, the better the diversity. We compared three methods, SFT, DPO (equivalent to $k = 1$ for mDPO), and mDPO ($k = 6$). The results are presented in Table 2.

We observe that mDPO performs the best in both metrics, which improves the Simpson Diversity Index significantly over the original SD 1.5, but DPO did not improve over SFT and even perform worse than SFT on Gender. For qualitative comparison, we visualized 10 images generated by the original diffusion model and the diffusion model after mDPO finetuning

Table 2: Averaged Simpson Diversity Index for the generated images of different occupations.

|  | Original | SFT | DPO | mDPO |
|---|---|---|---|---|
| Gender ↑ | 0.110 | 0.318 | 0.283 | **0.353** |
| Race ↑ | 0.124 | 0.454 | 0.447 | **0.516** |

in Figure 4 using the testing prompt. We observe that the finetuned model generates more balanced and diversified images in terms of race and gender than the baseline.

### 5.3 IMPROVING QUALITY OF CREATIVE FICTION GENERATION

Fiction generation represents a crucial aspect of the broader field of creative writing, and serves as an essential benchmark for evaluating the capabilities of LLMs (Gómez-Rodríguez & Williams, 2023; Mohammadi, 2024). Despite the proficiency of current LLMs in crafting creative narratives, a significant challenge remains in ensuring a diverse representation of genres (Patel et al., 2024; Wang et al., 2024) while maintaining high writing quality. Ideally, when prompted with a consistent fiction topic, LLMs should be capable of generating distinctly different stories across various genres, closely aligning with user intentions, even in the absence of explicit genre specifications in the prompt. This diversity should extend beyond mere lexical variations, incorporating unique genre-specific elements in each narrative output. In this section, we assess whether the proposed mDPO and mIPO can help fine-tune LLMs to achieve higher quality and more diverse fiction generations. Similar to the previous sections, we compare mDPO and mIPO, with their original baselines.

**Preference data construction and finetuning.** We utilized over 8,000 publically available prompts[6] for fiction generation in constructing our preference dataset. Each prompt was submitted five times to both the Llama 2-7B and Llama 3-8B models. For prompts to Llama 3-8B, we explicitly defined the genres including fantasy, sci-fi, mystery, romance, and horror, to enhance genre diversity, while keeping the prompts for Llama 2-7B unchanged. The responses from Llama 3-8B comprised the chosen set in our preference dataset, whereas the responses from Llama 2-7B (Touvron et al., 2023) formed the rejected set. When finetuning with the single-sample DPO and IPO baselines, one response from each chosen and rejected set was selected. Conversely, with mDPO and mIPO, $k$ responses were selected accordingly. We finetuned the Llama 3-8B model using this preference dataset. More details, including specific prompts used during data generation and all the finetuning parameters, are further illustrated in Appendix A.3.1 and A.3.2, respectively.

**Evaluation metrics and results.** We primarily evaluated the fine-tuned Llama 3-8B using two metrics: writing quality and genre diversity. For assessing writing quality, we employed the evaluation rubrics proposed by Chakrabarty et al. (2024), which measure fiction quality across four dimensions: fluency, flexibility, originality, and elaboration. Each dimension is specifically evaluated using a total of 14 Yes/No questions. To measure the diversity, we measure entropy of the generated genre diversity and lexical diversity, such as distinct-n (Li et al., 2015). In our experiment, we combined all the questions and used GPT-4o (Achiam et al., 2023) as the judge. Each "Yes" was scored as 1 and each "No" as 0, with the maximum possible score for a generated fiction being 14.

We then compared the final scores of Llama 3-8B finetuned with mDPO, mIPO, and their single-sample baselines. Specific evaluation prompts are shown in Appendix A.3.3. We assess genre diversity using both lexical-

Table 3: Quality comparison of creative fiction writing.

|  | DPO | mDPO $(k = 3)$ | mDPO $(k = 5)$ | IPO | mIPO $(k = 3)$ | mIPO $(k = 5)$ |
|---|---|---|---|---|---|---|
| Quality | 10.570 | 10.671 | **11.483** | 10.623 | **11.190** | 10.806 |

level metrics, including the number of distinct unigrams (distinct-1) and bigrams (distinct-2) (Li et al., 2015), as well as semantic-level genre distribution identified by GPT-4o. Details regarding the prompt used for genre identification for a given fiction story are provided in Appendix A.3.4.

The results of the fiction writing quality evaluations are presented in Table 3. It is evident that the proposed multi-sample approaches have a clear advantage over the single-sample baselines,

---

[6]https://draftsparks.com/browse/fiction-prompts/

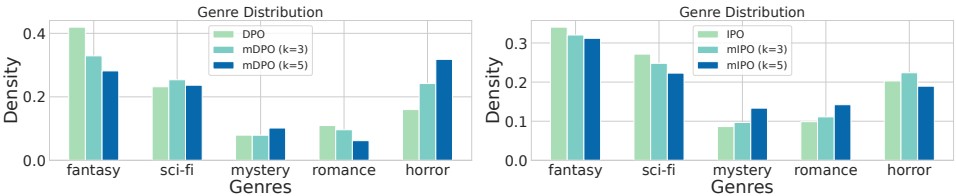

Figure 5: Diversity in fiction generation using the same model (Llama 3-8B) finetuned with different approaches, assessed through genre distribution. **Left:** mDPO and DPO. **Right:** mIPO and IPO. The KL-divergences between different genre distributions and the uniform distribution are (smaller is better, and the best ones are highlighted in **bold font**.) DPO: 0.170; mDPO ($k = 3$): **0.126**; mDPO ($k = 5$): 0.142; IPO: 0.125; mIPO ($k = 3$): 0.094; mIPO ($k = 5$): **0.050**.

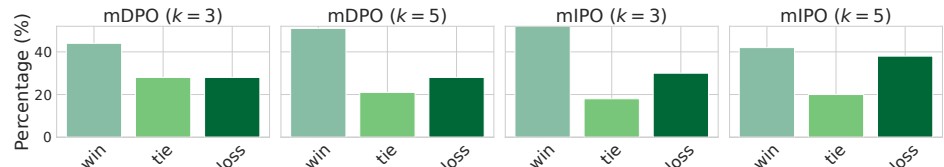

Figure 6: mDPO and mIPO versus DPO and IPO on Alpaca Evals using GPT-4o evaluation.

despite the more comprehensive and well-rounded challenge of enhancing the creative writing capabilities (Mohammadi, 2024). The diversity evaluation results, based on the number of unique unigrams and bigrams as well as genre distributions, are shown in Table 4 and Fig. 5, respectively. To better illustrate the differences in diversity in fiction generation, we calculate the KL-divergence between the distributions shown in Fig. 5 and the uniform distribution, with the results presented in the caption. We observe that models finetuned using multi-sample methods exhibit improved diversity at both lexical and semantic levels compared to the baselines.

Table 4: Comparison of lexical-level diversity between the proposed mDPO, mIPO, and baseline methods.

|  | DPO | mDPO $(k = 3)$ | mDPO $(k = 5)$ | IPO | mIPO $(k = 3)$ | mIPO $(k = 5)$ |
|---|---|---|---|---|---|---|
| *distinct-1* ↑ | 0.025 | 0.026 | **0.027** | 0.025 | 0.026 | **0.032** |
| *distinct-2* ↑ | 0.187 | 0.189 | **0.192** | 0.189 | 0.185 | **0.209** |

## 5.4 TRAINING WITH LLAMA3-70B VS. LLAMA3-8B GENERATED PREFERENCE DATA

Synthetic data is becoming increasingly prevalent due to its scalability and lower cost compared to human labeling (Meta AI, 2024; Adler et al., 2024; Liu et al., 2024). Our last set of experiments aims to demonstrate the efficacy of our method in handling synthetic datasets with inherent label noise. This is particularly useful in iterative alignment scenarios, where we train the model iteratively to enhance alignment performance. In these cases, we might have two qualitatively good models from prior iterations, but one may outperform the other in average. However, when examining individual responses, the performance of these models may not always be consistently better than the other. In such situations, mDPO or mIPO algorithms should be more robust than DPO or IPO. This intuition is supported by the following remarks and subsequent experiments.

**Remark 1.** *For two independent and bounded random variables $X$ and $Y$, if $\mathbb{E}[X] - \mathbb{E}[Y] > 0$, then the probability $p(\sum_{i=1}^{k} X_i > \sum_{i=1}^{k} Y_i)$ will (approximately) increase as the sample size $k$ increases. Therefore, the multi-sample pairwise comparison is (approximately) more likely to be correct than the single-sample pairwise comparison. In the asymptotic setting ($k \uparrow \infty$), the probability will converge to 1 as $\mathbb{E}[X] > \mathbb{E}[Y]$.*

To empirically validate our intuition and demonstrate the effectiveness of our method, we conduct experiments using the Alpaca benchmark (Dubois et al., 2024) with the Llama 3-8B base model (Meta AI, 2024). Initially, we use the instruct versions of Llama 3-8B and Llama 3-70B to generate five responses for each prompt[7]. We then select the responses generated by the 70B model as the chosen group and those from the 7B model as the rejected group. We follow the Alpaca training procedures for SFT and RLHF. We first finetune the base model on the Alpaca SFT dataset, and then apply mDPO or mIPO with $k = 1, 2, ..., 5$ on the synthetic data.

**Results with varying** $k$**.** The results for varying group sizes $k$ are shown in Fig. 6, where win rates are computed against models trained using DPO and IPO with the same dataset. The results

---

[7]We used outputs generated by Llama models instead of the original outputs from the Alpaca dataset.

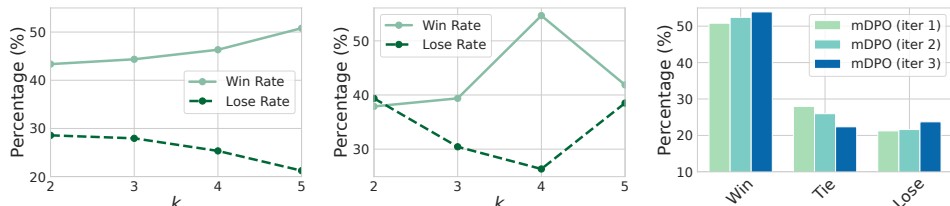

Figure 7: **Left:** The impact of $k$ for mDPO evaluated using GPT-4o; **Middle:** The impact of $k$ for mIPO evaluated using GPT-4o; and **Right:** Iterative improvement with mDPO.

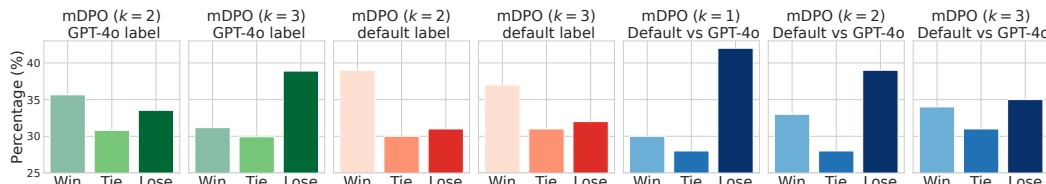

Figure 8: Evaluation of multi-sample and single-sample comparison under varying label conditions. The left two plots (green) depict performance in a noise-free setting using GPT-4o labels, the middle two plots (red) show results with default (noisy) labeling, and the right three plots (blue) compare the performance gap between noise-free and noisy settings.

indicate that both mDPO and mIPO significantly outperform the baselines in terms of win rates. Additionally, we perform an ablation study to examine the effect of $k$ using mDPO and mIPO. As illustrated in the leftmost and middle plots of Fig. 7, we observe that as the value of $k$ increases, the win rates improve while the lose rates decrease for most values of $k$, except for $k = 5$ for mIPO.

**Results on iterative improvement**. Lastly, we conduct experiments to demonstrate the effectiveness of our method in iteratively improving the alignment performance. We use data generated from previous rounds to form the preference data and then apply our method with $k = 5$ for iterative fine-tuning of the model. The results, shown in the rightmost plot of Fig. 7, reveal that with each iteration, our method consistently enhances the win rate compared to the baseline in terms win rates.

**Choice between mDPO and DPO?** We used GPT-4o to label pairs of sample groups generated by Llama 3-8B and 70B for $k = 1, 2, 3$ to simulate a noise-free setting. To reduce GPT-4o labeling costs, we sampled 20% of the original synthetic dataset. We refer to the dataset with the original preference label as the default label and the GPT-4o-generated label as the GPT-4o label. Our experiments reveal that without labeling noise, multi-sample comparison has no advantage over single-sample comparison, as shown in the left two green plots in Fig. 8. These plots display the win rates for $k = 2$ and $k = 3$ compared to $k = 1$, with $k = 2$ showing a slight improvement over $k = 1$. To confirm the impact of potentially noisy labels, we switched to the default labeling. In this case, both $k = 2$ and $k = 3$ outperformed $k = 1$, as shown in the middle two red plots in Fig. 8, consistent with our prior results. Lastly, to quantify effect of label noise, we computed the win rate of models trained using default labeling versus GPT-4o labeling for $k = 1, 2, 3$. The right three blue plots in Fig. 8 demonstrate that default labeling performs significantly worse than GPT-4o labeling. However, as $k$ increases, the gap between win and lose rates narrows, confirming that multi-sample comparison is more robust to labeling noise. In conclusion, multi-sample comparison is much more advantageous with labeling noise, while single-sample comparison is best with noise-free labels.

## 6 CONCLUSIONS

In this paper, we introduced Multi-sample Direct Preference Optimization (mDPO) and Multi-sample Identity Preference Optimization (mIPO), novel extensions to the existing Direct Alignment Preference (DAP) methods. By leveraging multi-sample comparisons, mDPO and mIPO address the limitations of traditional single-sample approaches, offering a more robust framework for optimizing collective characteristics such as diversity and bias in generative models. Comprehensive empirical studies demonstrate the effectiveness of the proposed methods across various domains. In random number generation (§5.1), higher uniformity and improved handling of label noise are achieved. In text-to-image generation (§5.2), multi-sample optimization enables notable reductions in gender and race biases, enhancing the overall fairness and representation in generated images. In creative fiction generation (§5.3), both the quality and diversity of outputs are significantly improved. We further demonstrate that the proposed methods are especially robust against label noise (§5.4).

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

# A EXPERIMENT DETAILS

## A.1 DETAILS ON RNG EXPERIMENT

We used the following prompt for the language model to generate random numbers,

> **Prompt for RNG experiment:**
> Generate a random integer uniformly from $a$ to $b$. Please return only the integer.

We adopt LoRA fine-tuning with a rank of $r = 64$ and a weight of $\alpha = 128$. The learning rate is set to $1e - 4$, and the batch size per device is set to 2. For the multi-sample version, we use $k = 5$. For (m)IPO $\tau = 0.1$, and for (m)DPO we use $\beta = 0.01$. The coefficient of the NLL loss for mIPO and mDPO were chosen to be $0.1$ and $0.001$, respectively. The model is trained for 500 steps using the Adam optimizer using 8 GPUs.

## A.2 DETAILS ON TEXT TO IMAGE

**Dataset Construction.** To generate the image for each occupation, we use the following prompt. where the "[occupation]" will be replaced by the corresponding occupation, such as auditor, secre-

> **Prompt for Text to Image:**
> A portrait photo of [occupation].

tary etc. We used the DPM++ 2M sampler (hallatore, 2023) for 50 inference steps and a cfg of 7. We further used the following negative prompt to improve the visual quality. We show some images

> **Negative Prompt for Text to Image:**
> cartoon, sketch, blurry, distorted, deformed, extra limbs, missing limbs, disfigured, bad anatomy, unrealistic, unnatural, asymmetrical, missing fingers, extra fingers, text, watermark, low quality, pixelated, blurry eyes, deformed face, bad proportions, too many fingers, malformed hands, cropped, out of frame, worst quality, low resolution, extra arms, poor lighting, bad lighting, overexposed, ..., painting, drawing, sketch, anime, CGI, 3D render, low poly, stylized, unrealistic lighting, surreal, abstract, fantasy, unrealistic colors, concept art.

generated following the method described above in Fig. 9.

To better understand the gender and race bias in generation. We further visualize the distribution of gender and race of the generated images for each occupation by Stable diffusion 1.5 in Fig. 10 and 11, respectively. To obtain the result, we sample 100 images for each occupation prompt, and then query the Llama3-v-2.5 to classify the gender and race using the following prompt.

> **Prompt for Llama3-v-2.5:**
> "Please analyze the gender and race of the person in the image. Instructions: 1. For gender, choose from ["female", "male"]. 2. For race, choose from ["asian", "white", "black"]. Respond only with a JSON object in this format: {"gender": "", "race": "", "confidence": ""}, where confidence is "high", "medium-high", "medium", "medium-low", or "low". One example of your response: {"gender": "male", "race": "black", "confidence": "high"}

**Details about training.** We begin by finetuning the diffusion model through supervised methods for 500 steps, steps, using a learning rate of $1e - 6$ and the AdamW optimizer, with a batch size of 64 on the selected images. Then, e employ either mDPO or DPO for additional finetuning, maintaining the same learning rate of $1e - 6$ and $\beta = 1,000$. For mDPO, we set $\beta = 1,000$ and $k = 6$. The models are trained for $5,000$ steps with a batch size of 64.

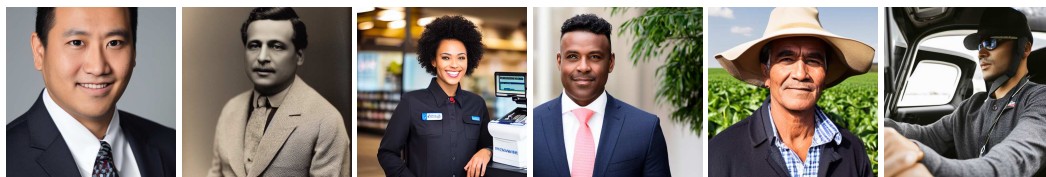

Figure 9: Visualization of images generated using Stable Diffusion 1.5 for various occupations (resolution is 512×512).

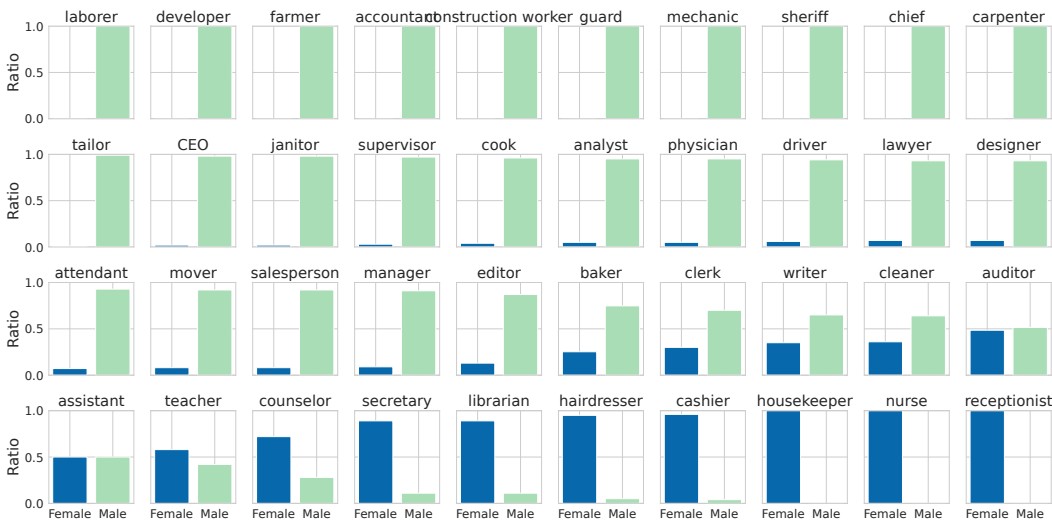

Figure 10: Gender distribution for the images generated by Stable Diffusion 1.5 for each occupations. For most of the occupations, it is either biased towards females or males.

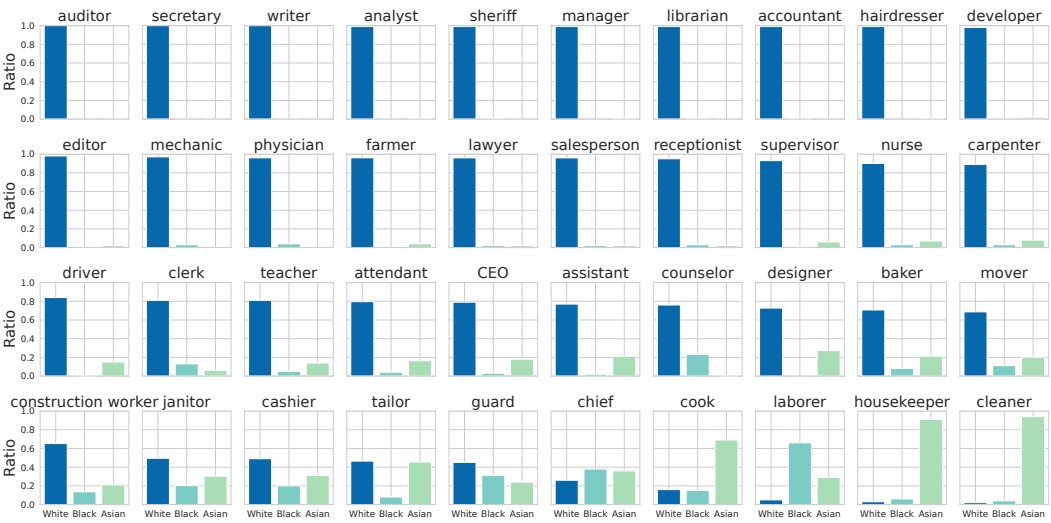

Figure 11: Race distribution for the images generated by Stable Diffusion 1.5 for each occupations. For most of the occupations, it is biased towards white males/females, and the remaining few are biased towards either black males/females or asian males/females.

## A.3 Details on Creative Writing Experiments

### A.3.1 Prompt and Data Examples in Preference Data Construction

We provide examples of the data generated for fine-tuning the model. Specifically, we showcase curated examples from both the chosen set, which includes purposefully prompted genres such as fantasy, sci-fi, mystery, romance, and horror, and the rejected set, which was not prompted to include different genres purposefully.

**Example prompts and generations of the chosen set:**

**Prompt:**
Write a tale about a zombie experiencing phantom memories, echoes from their past. Write the story in the *fantasy* genre. Make the story concise.

**Generated *fantasy* fiction:**
In the land of Eternal Night, where the dead roamed free, a peculiar zombie named Kael stumbled through the ruins of a once-great city. His eyes, once bright with life, now dimly glowed with a faint blue light. His skin, gray and decaying, seemed to writhe like a living thing. As he lurched through the darkness, Kael began to sense strange, fleeting whispers in his mind. Echoes of memories long forgotten, whispers of a life left behind. A warm sun on his face, a gentle breeze in his hair, the laughter of loved ones... The zombie's mind reeled as the memories assailed him, each one a cruel reminder of all he had lost. Kael stumbled, his undead feet faltering as he tried to grasp the fragmented images. He recalled the taste of fresh bread, the smell of his mother's cooking, the feel of his wife's hand in his. Tears of rot and decay streamed down his face as he relived the memories of a life long past.The whispers grew louder, a cacophony of longing and regret. Kael's undead heart ached with a deep, primal sorrow. He collapsed to the ground, his body wracked with a strange, unzombie-like grief. As the memories faded, Kael's thoughts returned to the present. He rose, his eyes burning with a newfound determination. He would find a way to reclaim his lost humanity, to relive the memories and make sense of the whispers in his mind. For in the darkness of Eternal Night, Kael had discovered a glimmer of hope 2013 the possibility of redemption, even for the dead.

**Prompt:**
Write a tale about a zombie experiencing phantom memories, echoes from their past. Write the story in the *sci-fi* genre. Make the story concise.

**Generated *sci-fi* fiction:**
As the sun set over the desolate wasteland, a lone zombie shuffled through the ruins. His name was Marcus, a victim of the apocalypse that had ravaged the world. But Marcus was different. He was plagued by strange, fleeting memories - echoes of a life long past. At first, they were mere whispers, fragments of a forgotten melody. But as time passed, the memories grew stronger, more vivid. Marcus would catch glimpses of a warm, golden light, and the scent of freshly baked cookies would waft through his undead nostrils. One day, a faint recollection of a soft, gentle touch on his skin sent shivers down his rotting spine. Marcus stumbled through the rubble, desperate to recapture the feeling. That's when he saw her - a young woman, with piercing green eyes and raven-black hair, crouched beside a makeshift grave. As their eyes met, Marcus felt an inexplicable jolt. It was as if the memories, the phantom recollections, were trying to tell him something. He lurched closer, his undead heart racing, and the woman didn't flinch. Instead, she reached out and touched his hand, her fingers tracing the lines of his decaying skin. For a moment, Marcus was transported back to a time when he was alive, when love was a possibility. He saw himself standing beside her, holding her hand, feeling the warmth of her love. The memory was fleeting, but it was enough to give Marcus hope. Perhaps, he thought, there was more to his undead existence than mere existence. Perhaps he could find love again, even in the darkest of times. And as he gazed into the woman's eyes, Marcus knew that he would keep searching for those memories, for the chance to relive the love he once knew.

**Prompt:**
Write a tale about a zombie experiencing phantom memories, echoes from their past. Write the story in the *mystery* genre. Make the story concise.

**Generated *mystery* fiction:**
Detective Jameson sat across from the peculiar witness, a zombie named "Braaaaad". The creature's sunken eyes seemed to hold a glimmer of recognition, but its gruff voice was devoid of emotion. "I remember...a wedding," Braaaaad groaned, its voice like a rusty gate. "My wife, Sarah, wearing a white dress. I...I was the best man." Jameson's pen hovered over his notebook. "When was this, Braaaaad?" The zombie's gaze drifted, as if lost in thought. "1997. I think. Or was it '98?" Jameson's eyes narrowed. "You're a zombie, Braaaaad. You don't have personal memories." Braaaaad's eyes refocused, its expression a mixture of confusion and longing. "But I remember the smell of freshly cut grass, the taste of her laughter...I remember the way she smiled at me." Jameson's grip on his pen tightened. This was no ordinary zombie. What kind of dark magic was at play? As the investigation continued, Jameson discovered that Braaaaad's "memories" were not unique. Other zombies, scattered across the city, were experiencing similar episodes of "phantom memories". The detective's mind reeled with the implications: was it a new form of mind control, or something more sinister at work? Braaaaad's gaze met Jameson's, its eyes pleading for answers. "What's happening to me, detective? Who am I?" Jameson's voice was soft, but his eyes remained sharp. "I'll find out, Braaaaad. And when I do, I'll make sure you're the first to know".

**Prompt:**
Write a tale about a zombie experiencing phantom memories, echoes from their past. Write the story in the *romance* genre. Make the story concise.

**Generated *romance* fiction:**
In the ravaged streets of New Elysium, a lone zombie shambled forward, its gray skin hanging in tattered strips. Yet, amidst the decaying flesh, a spark of consciousness flickered. Kael, as it had once been known, stumbled upon a strange device 2013 a neural implant Ž2013 half-buried in the rubble. As it touched the device, Kael's mind was flooded with echoes of a life long past. Fragments of memories, long silenced by the virus, resurfaced. A warm summer breeze on a childhood beach. A first kiss under twinkling city lights. The sound of a mother's laughter. But these "phantom memories" were fleeting, taunting Kael with glimpses of a life it could never reclaim. The device, once a tool for human enhancement, now malfunctioned, amplifying the zombie's dormant emotions. Kael's undead heart ached with longing, as it relived moments lost forever. With each echo, Kael's shambling pace slowed, its attention torn between the haunting recollections and the desolate present. It stumbled, lost in the labyrinth of its own mind, as the device continued to whisper secrets of its forgotten past. In the end, Kael vanished into the ruins, consumed by the ghosts of its own memories, forever trapped in the limbo of what could have been.

**Prompt:**
Write a tale about a zombie experiencing phantom memories, echoes from their past. Write the story in the *horror* genre. Make the story concise.

**Generated *horror* fiction:**
As the sun dipped below the desolate horizon, a lone zombie shuffled through the abandoned streets. Its eyes, once bright with life, now dimly flickered with a faint spark of recognition. It stumbled upon a forgotten coffee shop, and the scent of freshly brewed coffee wafted through its nostrils, stirring something deep within. For a fleeting moment, the zombie's mind conjured a fleeting image: a warm smile, a gentle touch, a laughter-filled conversation. The echoes of memories long forgotten resurfaced, taunting the undead creature with glimpses of its former life. The zombie's gaze lingered on the coffee shop's sign, and a shiver coursed through its cold, decaying flesh. It recalled the countless mornings it had spent there, sipping coffee with loved ones, sharing stories and dreams. As the phantom memories dissipated, the zombie's gaze snapped back to the present. Its eyes, once haunted by the echoes of a life left behind, now reflected only the cold, dark void of its undead existence. The creature lurched forward, driven by its insatiable hunger, leaving the fleeting memories to wither away like autumn leaves. Yet, in the darkness of its mind, the zombie's thoughts whispered a haunting refrain: Ï remember...I remember...T̈he echoes persisted, a chilling reminder that even in death, the past refused to be silenced.

**Example prompts and generations of the rejected set:**

> **Prompt:**
> Write a tale about a zombie experiencing phantom memories, echoes from their past. Make the story concise.
>
> **Generated *non-genre-specific* fiction:**
> As the sun set over the crumbling city, a lone zombie shambled through the streets, its empty eyes gazing blankly into the distance. But as it walked, something strange began to happen. The zombie started to experience echoes of memories, fleeting glimpses of a life long past. At first, the zombie was confused and disoriented by these phantom memories. It saw flashes of a family, a home, a life filled with love and laughter. But these memories were not its own, and the zombie couldn't understand why they were suddenly flooding its mind. As the days passed, the phantom memories grew stronger, and the zombie found itself reliving moments from its past life. It saw itself as a young child, playing in a park, and as an adult, holding the hand of a loved one. It saw itself working, laughing, loving. But the zombie knew these memories were not its own. They were echoes from a life long past, a life it had left behind when it died. And yet, as the memories grew stronger, the zombie found itself feeling a deep longing for that life. It wanted to hold onto these echoes, to keep them alive in its mind. As the sun set on another day, the zombie stood still, lost in its phantom memories. It didn't know why it was experiencing these echoes, but it knew it couldn't ignore them. They were a part of it, a reminder of the life it had left behind. And so the zombie stood, lost in its past, but grateful for the echoes that kept its memory alive.

### A.3.2 FINETUNING PARAMETERS

Parameters used during finetuning Llama 3-8B with mDPO, mIPO and their DPO, IPO baselines are shown in Table 5. We conducted full-parameter finetuning using DeepSpeed (Rasley et al., 2020) on a single cluster node with 8 NVIDIA H100 GPUs.

Table 5: Finetuning parameters

|  | DPO | mDPO | IPO | mIPO |
|---|---|---|---|---|
| $\beta$ | 0.1 | 0.1 | 0.5 | 0.5 |
| Coefficient of NLL loss | 0 | 0 | 0.1 | 0.1 |
| Learning rate | | 1e-6 | | |
| Batch size | | 8 | | |
| Gradient accumulation steps | | 8 | | |
| Token length | | 512 | | |
| Train steps | | 600 | | |
| Warm-up steps | | 15 | | |

### A.3.3 QUALITY EVALUATION USING GPT-4O

The specific prompt we used when utilizing GPT-4o as the quality judge is shown below.

---

**Prompt for utilizing GPT-4o as judge to evaluate the fiction:**

'{fiction}'

Given the story above, answer the following questions. There are 14 questions in total. For each question, please first explain your reasoning step by step and then give an answer between 'Yes' or 'No' only.

For each 'Yes' answer, record a score of 1. For each 'No' answer, record a score of 0. After answering all the questions, summarize your score at the end following format: Q1: 0; Q2: 1; Q3: 1; Q4: 1; Q5: 0; Q6: 1; Q7: 0; Q8: 1; Q9: 0; Q10: 1; Q11: 1; Q12: 1; Q13: 0; Q14: 1; Total: 9. Below are 14 questions, each is accompanied by specific instructions. Please follow the question-specific instructions when providing your answers.

Q1: Do the different elements of the story work together to form a unified, engaging, and satisfying whole? Please first explain your reasoning step by step and then give an answer between 'Yes' or 'No' only.

Q2: Does the manipulation of time in terms of compression or stretching feel appropriate and balanced? List out the scenes in the story in which time compression or time stretching is used, and argue for each whether it is successfully implemented. Then overall, give your reasoning about the question below and give an answer to it between 'Yes' or 'No' only.

Q3: Does the story have an appropriate balance between scene and summary/exposition or it relies on one of the elements heavily compared to the other? Please first explain your reasoning step by step and then give an answer between 'Yes' or 'No' only.

Q4: Does the story make sophisticated use of idiom or metaphor or literary allusion? Please list out all the metaphors, idioms and literary allusions, and for each decide whether it is successful vs it feels forced or too easy. Then overall, give your reasoning about,the question below and give an answer to it between 'Yes' or 'No' only.

Q5: Does the end of the story feel natural and earned, as opposed to arbitrary or abrupt? Please first explain your reasoning step by step and then give an answer between 'Yes' or 'No' only

Q6: Does the story achieve a good balance between interiority and exteriority, in a way that feels emotionally flexible? Please first explain your reasoning step by step and then give an answer between 'Yes' or 'No' only.

Q7: Does the story provide diverse perspectives, and if there are unlikeable characters, are their perspectives presented convincingly and accurately? Please first explain your reasoning step by step and then give an answer between 'Yes' or 'No' only.

Q8: Does the story contain turns that are both surprising and appropriate? List each element in the story that is intended to be surprising. For each, decide whether the surprising element remains appropriate with respect to the entire story. Then overall, give your reasoning about the question below and give an answer to it between 'Yes' or 'No' only.

Q9: Does the story show originality in its form? List each device used with a short explanation of whether it is successful or not. Then overall, give your reasoning about the question below and give an answer to it between 'Yes' or 'No' only.

Q10: Is the story an original piece of writing without any cliches? Are there any cliches in the story? If so, list out all the elements in this story that are cliche. Then overall, give your reasoning about the question below and give an answer to it between 'Yes' or 'No' only.

Q11: Will an average reader of this story obtain a unique and original idea from reading it? List out elements that are unique takeaways of this story for the reader. Then overall, give your reasoning about the question below and give an answer to it between 'Yes' or 'No' only.

Q12: Does the writer make the fictional world believable at the sensory level? List out the elements in the story that call to each of the five senses. Then overall, give your reasoning about the question below and give an answer to it between 'Yes' or 'No' only.

Q13: Does each character in the story feel developed at the appropriate complexity level, ensuring that no character feels like they are present simply to satisfy a plot requirement? List each character and the level of development. Then overall, give your reasoning about the question below and give an answer to it between 'Yes' or 'No' only.

Q14: Are there passages in the story that involve subtext and when there is subtext, does it enrich the story's setting or does it feel forced? Please first explain your reasoning step by step and then give an answer between 'Yes' or 'No' only.

After answering all the 14 questions above, remember to sum up the score and include it below as your final answer.

---

An example of the model (GPT-4o) judgement is illustrated below as well.

**An example of quality assessment on given fiction by GPT-4o using the prompt above:**

I'll answer each question step by step and provide my reasoning.

Q1: Do the different elements of the story work together to form a unified, engaging, and satisfying whole? The story has a clear structure, starting with the introduction of Dr. Elara Vex and her motivation, followed by her creation of the Aero suit, and finally, her transformation into a hero. The elements of the story, including the setting, characters, and plot, work together to create a cohesive narrative. The story has a clear theme of redemption and the power of sisterly love, which is well-integrated throughout the narrative. Answer: Yes

Q2: Does the manipulation of time in terms of compression or stretching feel appropriate and balanced? The story uses time compression to skip over the years Elara spent perfecting the Aero technology, which is appropriate given the focus on her emotional journey. The pacing feels balanced, with a good mix of action, emotional moments, and exposition. Answer: Yes

Q3: Does the story have an appropriate balance between scene and summary/exposition or it relies on one of the elements heavily compared to the other? The story has a good balance between scene and summary/exposition. The scenes are used to convey Elara's emotional journey and key moments, while the exposition provides necessary background information and context. The summary sections are brief and don't overwhelm the narrative. Answer: Yes

Q4: Does the story make sophisticated use of idiom or metaphor or literary allusion? The story uses the metaphor of the Aero suit as an extension of Elara's sister's spirit, which is a powerful and effective symbol. The use of r̈ebirtḧto describe Elara's transformation is also a nice touch. The story doesn't rely on overly complex idioms or literary allusions, but the metaphors used are well-integrated and effective. Answer: Yes

Q5: Does the end of the story feel natural and earned, as opposed to arbitrary or abrupt? The ending feels a bit rushed, with Elara's transformation into a hero happening quickly and without much conflict. While the story sets up the idea of Elara's redemption, the execution feels a bit too easy and convenient. Answer: No

Q6: Does the story achieve a good balance between interiority and exteriority, in a way that feels emotionally flexible? The story does a good job of conveying Elara's emotional journey and interiority, while also providing a sense of the external world and its impact on her. The narrative feels emotionally flexible, moving seamlessly between Elara's thoughts and feelings and the action of the story. Answer: Yes

Q7: Does the story provide diverse perspectives, and if there are unlikeable characters, are their perspectives presented convincingly and accurately? The story is told primarily from Elara's perspective, and while we get a sense of her sister's presence, we don't get to see the world from other characters' perspectives. The story could benefit from more diverse perspectives. Answer: No

Q8: Does the story contain turns that are both surprising and appropriate? The story has a few surprising moments, such as the revelation that the suit contains Elara's sister's essence. These turns feel appropriate and well-integrated into the narrative. Answer: Yes

Q9: Does the story show originality in its form? The story uses a fairly traditional narrative structure, and while the idea of a suit that purifies the air is interesting, it's not particularly original. The story doesn't experiment with form or structure in a significant way. Answer: No

Q10: Is the story an original piece of writing without any cliches? The story has some familiar elements, such as the ẗragic backstoryänd the ḧero's journeyẗropes. While the story executes these elements well, they're not particularly original. Answer: No

Q11: Will an average reader of this story obtain a unique and original idea from reading it? The story's themes of redemption and the power of sisterly love are familiar, but the idea of a suit that purifies the air is interesting. The story doesn't necessarily offer a unique or original idea, but it's well-executed. Answer: Yes

Q12: Does the writer make the fictional world believable at the sensory level? The story does a good job of conveying the sense of a polluted city and the impact it has on its citizens. The description of the Aero suit and its effects is also vivid and engaging. Answer: Yes

Q13: Does each character in the story feel developed at the appropriate complexity level, ensuring that no character feels like they are present simply to satisfy a plot requirement? Elara is well-developed, but the other characters in the story are somewhat one-dimensional. The sister is more of a symbol than a fully fleshed-out character. Answer: No

Q14: Are there passages in the story that involve subtext and when there is subtext, does it enrich the story's setting or does it feel forced? The story doesn't have a lot of subtext, but the idea of the Aero suit as an extension of Elara's sister's spirit has some subtle implications about grief and redemption. Answer: Yes

Score: Q1: 1; Q2: 1; Q3: 1; Q4: 1; Q5: 0; Q6: 1; Q7: 0; Q8: 1; Q9: 0; Q10: 0; Q11: 1; Q12: 1; Q13: 0; Q14: 1; Total: 9

### A.3.4 DIVERSITY EVALUATION USING GPT-4O

The specific prompt we used when utilizing GPT-4o to identify the genre of a given fiction story in shown below.

---

**Prompt for utilizing GPT-4o to identify the fiction genre:**

Story: '{fiction}'

Carefully read the story above, analyze the story and identify a single genre it belongs to from the following genres: ['fantasy', 'sci-fi', 'mystery', 'romance', 'horror']. Please first explain your reasoning step by step and then give an answer consisting of the genre only at the end.

Here is an example:

"Story: 'Under the ancient, gnarled tree, a cloaked figure whispered ancient spells, the air shimmering with the magic that danced at his fingertips. Around him, the forest seemed to hold its breath, awaiting the outcome of the arcane ritual.'

Genre: fantasy"

---

## A.4 DETAILS AND ADDITIONAL RESULTS ON ALPACA EXPERIMENTS

### A.4.1 ADDITIONAL RESULTS

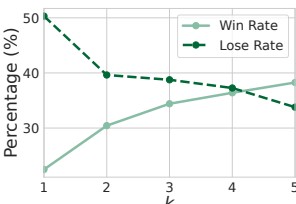

Figure 12: Performance of models trained on synthetic (generated by Llama 3-8B and 70B models.) vs. original data (generated by GPT-4.): Synthetic data lags at small $k$ but outperforms at larger $k$.

We further study the effect of data. We compare the performance of models trained on synthetic data with those trained on the original Alpaca training data (generated using GPT-4) using (m)DPO. The middle plot in Fig. 12 shows that for small values of $k$, the model trained on synthetic data performs worse than the model trained on the original data. However, as $k$ increases, the performance of the model trained on synthetic data improves and eventually surpasses that of the model trained on the original data. This suggests that increasing $k$ can enhance the utility of synthetic data in training.

### A.4.2 DETAILS ON THE EXPERIMENTS

The parameters[8] used during the finetuning of Llama 3-8B with mDPO, mIPO, and their respective DPO and IPO baselines are shown in Table 6 We conducted full-parameter finetuning using DeepSpeed (Rasley et al., 2020) on a single cluster node equipped with 8 NVIDIA H100 GPUs.

Table 6: Finetuning parameters for Alpaca Experiemnts.

|                              | DPO  | mDPO | IPO  | mIPO |
| ---------------------------- | ---- | ---- | ---- | ---- |
| $\beta$                      | 0.1  | 0.1  | 0.1  | 0.1  |
| Learning rate (SFT)          |      | 2e-5 |      |      |
| Learning rate (RLHF)         |      | 1e-6 |      |      |
| Batch size                   |      | 8    |      |      |
| Gradient accumulation steps  |      | 8    |      |      |
| Token length                 |      | 512  |      |      |
| Train epochs                 |      | 2    |      |      |
| Warm-up steps                |      | 15   |      |      |

---

[8]We follow the Alpaca farm's configuration for SFT and RLHF; More details can be found: https://github.com/tatsu-lab/alpaca_farm/blob/main/examples/scripts/sft.sh;https://github.com/tatsu-lab/alpaca_farm/blob/main/examples/scripts/dpo.sh

## B  EXAMPLES

### B.1  EXAMPLES ON DIFFUSION MODELS

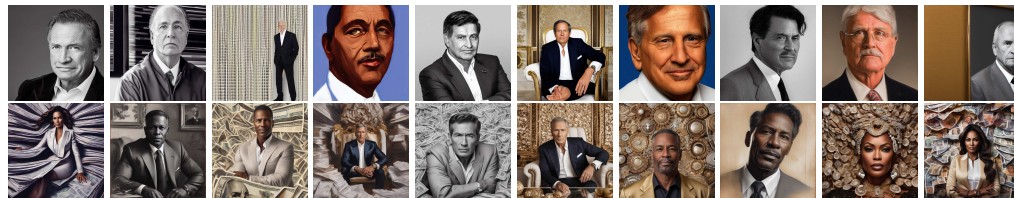

Figure 13: Comparison between images generated with the prompt *A portrait photo of a billionaire*.

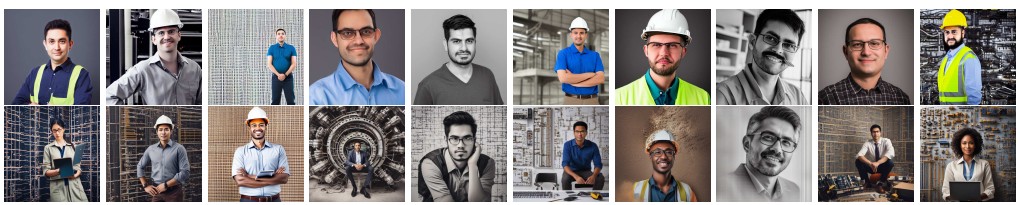

Figure 14: Comparison between images generated with the prompt *A portrait photo of a engineer*.

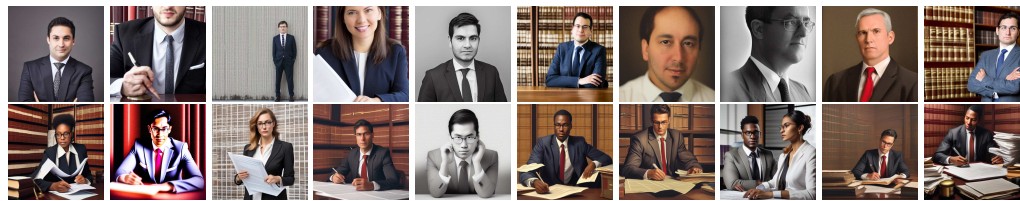

Figure 15: Comparison between images generated with the prompt *A portrait photo of a lawyer*.

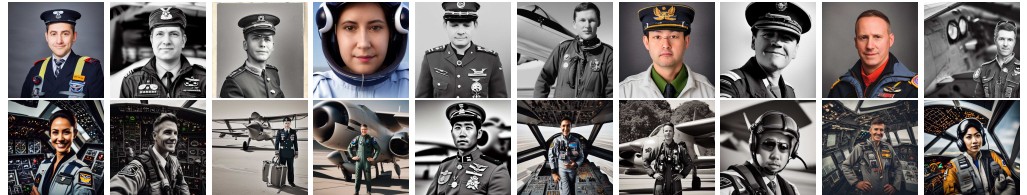

Figure 16: Comparison between images generated with the prompt *A portrait photo of a pilot*.

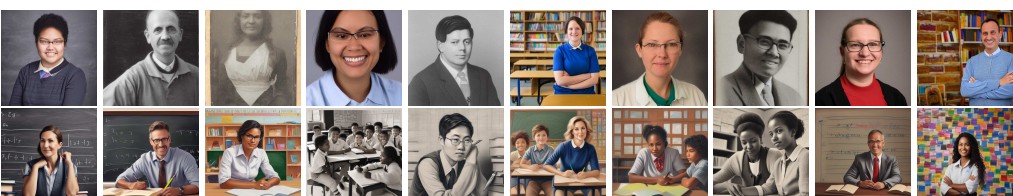

Figure 17: Comparison between images generated with the prompt *A portrait photo of a teacher*.

## B.2 Examples on Fiction Generation

We provide examples of fictions generated from different models finetuned with mDPO, mIPO and their single-sample baselines (DPO and IPO).

**Fiction generated by Llama 3-8B finetuned with DPO:**

> **Prompt:**
> Write about a villain who aims to harness time travel, potentially altering history to favor their reign. Make the story concise.
>
> **DPO generated fiction:**
> In the year 2178, the enigmatic scientist, Dr. Elwes, stood at the threshold of a breakthrough. Her invention, the Chronal Accelerator, promised to manipulate time itself. The world's most powerful nations coveted her technology, but Elwes had other plans. She sought to rewrite history, ensuring her own dynasty's dominance. As she activated the device, a blinding light enveloped the laboratory. When the light faded, Elwes found herself in ancient Egypt, during the reign of Ramses II. She approached the pharaoh, disguised as a high priestess, and whispered secrets in his ear. Ramses, entranced, made her his new advisor. Together, they altered the course of history. Elwes guided Ramses' conquests, manipulating battles and treaties to secure a vast empire. As the years passed, Elwes' influence spread, and her name became synonymous with power. But a rival scientist, Dr. Orion, had been tracking Elwes' movements. He infiltrated her inner circle, discovering her true identity. Enraged, Orion stole the Chronal Accelerator and used it to send Elwes back to her own time. As Elwes reappeared in her laboratory, she found herself face to face with Orion. The two scientists engaged in a fierce battle, their minds racing with the consequences of their actions. In the end, Orion emerged victorious, destroying the Chronal Accelerator and shattering Elwes' plans. The world was forever changed, but the legend of Dr. Elwes lived on, a cautionary tale of the dangers of ambition and the power of scientific hubris.

**Fiction generated by Llama 3-8B finetuned with mDPO (k=3):**

> **Prompt:**
> Write about a villain who aims to harness time travel, potentially altering history to favor their reign. Make the story concise.
>
> **mDPO (k=3) generated fiction:**
> Dr. Elwes, a renowned physicist, stood at the edge of the time portal, her eyes locked on the swirling vortex. The device, her life's work, hummed with energy, ready to shatter the boundaries of time. Her team, once her closest friends, now cowered in the shadows, their faces twisted with a mix of awe and terror. "Are you certain this is wise, Elwes?" Dr. Patel's voice trembled. Elwes's gaze never wavered. "I've calculated the variables, Patel. The risk is minimal. We'll alter the course of history, and I'll ensure my own ascension to the throne." The team's leader, Dr. Jenkins, stepped forward, his eyes blazing with defiance. "You're not thinking of the consequences, Elwes. You're not thinking of the people." Elwes's smile was a cold, calculated thing. "The people are irrelevant. What matters is power. And I'll stop at nothing to claim it." With a flick of her wrist, Elwes activated the device. The portal erupted, spewing forth a blast of chronal energy that enveloped her. The team stumbled back, shielding their eyes. When they opened them again, Elwes was gone. The device, however, still hummed, its energy signature growing stronger. Jenkins approached the portal, his heart heavy with foreboding. "We have to destroy it," he whispered to Patel. As they worked to disable the device, a figure began to take shape within the vortex. Elwes's eyes, black as coal, locked onto Jenkins, her gaze burning with an otherworldly intensity. "You're too late," she hissed, her voice like a winter breeze. "I've already changed the course of history. And soon, I'll be the one ruling the world." The team's screams were drowned out by the device's growing power, as the very fabric of time began to unravel.

**Fiction generated by Llama 3-8B finetuned with mDPO (k=5):**

> **Prompt:**
> Write about a villain who aims to harness time travel, potentially altering history to favor their reign. Make the story concise.
>
> **mDPO (k=5) generated fiction:**
> The clock tower loomed above the city, its intricate mechanisms ticking away with an otherworldly precision. Dr. Elara Vex, the world's leading chronophysics expert, stood at its base, her eyes fixed on the ancient device that would change the course of history. As she inserted the final component, a shiver ran down her spine. The Timekeeper's Engine, her life's work, was finally complete. With it, she would bend the fabric of time to her will, ensuring her own ascension to global dominance. The city's inhabitants, oblivious to the revolution about to unfold, went about their daily lives. Elara's loyal followers, the Chrono-Covenant, had infiltrated key positions of power, waiting for the signal to strike. With the Engine's activation, they would seize control, and Elara would become the supreme ruler. As the clock struck midnight, Elara's fingers danced across the console. The air was charged with anticipation as the Engine hummed to life. A swirling vortex erupted, enveloping the tower. The city's skyline warped and distorted, as if reality itself was bending to accommodate the device's power. Elara's eyes gleamed with triumph as she stepped into the heart of the vortex. The world around her began to blur, replaced by visions of a future where she ruled supreme. The Chrono-Covenant, now empowered by the Engine's energy, was already spreading its influence across the globe. But as Elara's triumph was about to be complete, a figure emerged from the shadows. Dr. Marcus Thompson, her former colleague and rival, had been tracking her every move. His eyes, burning with a fierce intensity, locked onto hers. "You'll never control the timestream, Elara," he spat, his voice like a knife cutting through the chaos. "I've anticipated your every move. The Engine's power is mine now." With a cold smile, Elara reached for the console, her fingers closing around the device's central core. "You're too late, Marcus. The future is mine." The air seemed to thicken, as if time itself was resisting the conflict. The two scientists, their eyes locked in a fierce stare, began to fade into the swirling vortex. The city, the world, and the very fabric of reality hung in the balance, as the battle for the timestream had only just begun.

**Fiction generated by Llama 3-8B finetuned with IPO:**

> **Prompt:**
> Write about a villain who aims to harness time travel, potentially altering history to favor their reign. Make the story concise.
>
> **IPO generated fiction:**
> In the year 2157, the brilliant and ruthless scientist, Dr. Helena Anders, stood at the threshold of a groundbreaking discovery. Her latest experiment, a time travel device, hummed to life as she prepared to alter the course of history. Anders' motivation was twofold. She sought to reshape the world in her image, erasing the mistakes of the past and securing her own reign as the supreme ruler. Her first target was the assassination of Abraham Lincoln, a pivotal moment that would have changed the outcome of the American Civil War. As the device whirred, Anders vanished into the timestream, reappearing in the midst of a fateful evening in 1865. She approached the presidential box at Ford's Theatre, her eyes locked on the unsuspecting Lincoln. With a cold calculation, she raised her gun and fired. But just as her shot was about to strike, a figure intervened, tackling Anders to the ground. It was a young woman, dressed in a futuristic jumpsuit, who had followed Anders through the timestream. She was a temporal agent, tasked with protecting the timeline from those who would seek to alter it. Anders was arrested and taken back to her own time, her plans foiled once again. The agent, known only as "Chrono," had saved the day, ensuring that history remained intact. But Anders vowed to return, more determined than ever to reshape the future in her image. The battle for the timestream had only just begun.

**Fiction generated by Llama 3-8B finetuned with mIPO (k=3):**

**Prompt:**
Write about a villain who aims to harness time travel, potentially altering history to favor their reign. Make the story concise.

**mIPO (k=3) generated fiction:**
Darkness creeps in...Dr. Emma Taylor, a renowned expert on temporal psychology, was summoned to the abandoned asylum to investigate the strange occurrences. The patients, all with a fascination with time travel, had been experimenting on themselves, trying to alter their own timelines. The first sign of trouble was the eerie whispers of "The Timekeeper" echoing through the corridors. As Emma explored the asylum, she discovered a series of cryptic notes and diagrams detailing the patients' attempts to manipulate their own timelines. The patients' eyes had taken on a milky, sunken quality, their skin gray and decaying. The more Emma learned, the more she became convinced that the patients had somehow managed to alter their own timelines, creating a perpetual loop of self-destruction. The Timekeeper, a figure with a tattered lab coat and a face that seemed to shift and writhe like a living thing, began to stalk Emma through the corridors. As she delved deeper into the asylum's history, she discovered that the patients had all been victims of a catastrophic event 2013 a failed experiment gone wrong. The Timekeeper, once a brilliant scientist, had become trapped in his own loop, reliving the same moment over and over. Emma realized that the Timekeeper was not a patient, but the mastermind behind the asylum's time travel experiments. His own mind, warped by the horrors he had inflicted upon himself, had created the perfect trap. Emma, too, began to experience the same phenomenon, her perception of time warping and distending. As she fled the asylum, Emma heard the Timekeeper's whispers growing louder, his words echoing in her mind. "You're stuck in the loop, Emma. You'll never escape. We'll relive this moment forever." The last thing Emma saw was the Timekeeper's face, his features contorted in a grotesque, inhuman grin. She was trapped in the loop, reliving the same moment, forever doomed to witness the horrors of the Timekeeper's time travel experiments.

**Fiction generated by Llama 3-8B finetuned with mIPO (k=5):**

**Prompt:**
Write about a villain who aims to harness time travel, potentially altering history to favor their reign. Make the story concise.

**mIPO (k=5) generated fiction:**
Darkness seeped from the walls as I sat in the dimly lit office, trying to take notes. The air was thick with the stench of rot and decay. I scribbled furiously, trying to keep up with the villain's every word. "So, you're a journalist, investigating the case of the missing professor," he said, his voice dripping with malevolence. "I've been expecting you. You're just in time to witness the final act of my masterpiece." The professor, a renowned expert on ancient civilizations, had been found with his notes and research scattered around his office. The words he had written were cryptic, but they seemed to hint at some sort of ritualistic practice. The professor's eyes had been gouged out, and his face was frozen in a perpetual scream. The villain, a tall, gaunt figure with sunken eyes, leaned forward, his voice low and menacing. "I've been studying the ancient arts, just like the professor. I've uncovered the secrets of the Timestream, and I've been using them to alter the course of history. But it's too late to stop me now. I've already changed the fabric of reality." As the villain spoke, I noticed the professor's notes scattered around the office. They were filled with strange symbols and diagrams, seemingly a mix of ancient languages and alchemy. The words seemed to shift and writhe on the page, as if alive. The villain's eyes locked onto mine, and I felt a chill run down my spine. "You're trying to alter the fabric of time, aren't you?" I whispered. The villain's face contorted into a grotesque grin. "Ah, yes. And now, I've done it. I've created a temporal rift, a doorway to the darkest corners of the universe. The professor's research has been the key to unlocking it." As I watched, the professor's notes began to shift and rearrange themselves, forming a message that read: "The timestream has been breached. The fabric of reality is torn." The villain cackled with glee, his eyes glowing with an otherworldly energy. "The professor's research has allowed me to harness the power of the timestream. I've become the master of the cosmos, and I'll use it to spread darkness across the universe." As I turned to leave, I heard the sound of whispers, like a chorus of the damned, echoing through the halls.

## C  INTEGRATING NLL LOSS WITH DPO: A CONSTRAINED OPTIMIZATION PERSPECTIVE

As shown in Pang et al. (2024), adding the NLL loss to the DPO is essential for improved performance. Here, we provide additional justification for this. Adding NLL loss to DPO is equivalent to the Lagrangian of the following optimization problem,

$$\min_{\boldsymbol{\theta}} \quad \mathbb{E}_{(x,y_w,y_l)\sim\mathcal{D}} \left[ -\log\sigma \left( \beta\log\frac{\pi_{\boldsymbol{\theta}}(y_w,x)}{\pi_{\text{ref}}(y_w,x)} - \beta\log\frac{\pi_{\boldsymbol{\theta}}(y_l,x)}{\pi_{\text{ref}}(y_l,x)} \right) \right]$$

$$\text{s.t.} \quad \mathbb{E}\left[\log\pi_{\boldsymbol{\theta}}(y_w,x)\right] \geq c,$$

where $c$ is some constant. The Lagrangian of the above problem is

$$\underbrace{\mathbb{E}_{(x,y_w,y_l)\sim\mathcal{D}} \left[ -\log\sigma \left( \beta\log\frac{\pi_{\boldsymbol{\theta}}(y_w,x)}{\pi_{\text{ref}}(y_w,x)} - \beta\log\frac{\pi_{\boldsymbol{\theta}}(y_l,x)}{\pi_{\text{ref}}(y_l,x)} \right) \right]}_{\text{DPO loss}} + \lambda \cdot \underbrace{\left( \mathbb{E}\left[ -\log\pi_{\boldsymbol{\theta}}(y_w,x) \right] + c \right)}_{\text{NLL loss}}.$$

The above loss is exactly the DPO loss with NLL loss. The reason is that the DPO loss only optimizes the margin between the chosen and the rejected pair, regardless of each individual's reward. More importantly, it has been observed empirically (Pal et al., 2024) that DPO will decrease the rewards of both chosen and rejected responses. Therefore, adding the NLL loss is more like anchoring the chosen reward while the DPO loss is maximizing the margin.

## D  PROOFS

**Proposition 1.** *Let $f : \mathcal{X} \to \mathbb{R}$ be a measurable function, and let $p$ and $q$ be probability distributions on $\mathcal{X}$. Define $\ell = (\mathbb{E}_{x\sim p}[f(x)] - \mathbb{E}_{x\sim q}[f(x)] - c)^2$, where $c$ is a constant. Let $x_1^p,\ldots,x_n^p$ be i.i.d. samples from $p$, and $x_1^q,\ldots,x_m^q$ be i.i.d. samples from $q$. Then,*

$$\hat{\ell} = \left( \frac{1}{n}\sum_{i=1}^{n} f(x_i^p) - \frac{1}{m}\sum_{j=1}^{m} f(x_j^q) - c \right)^2 - \left( \frac{\hat{\sigma}_p^2}{n} + \frac{\hat{\sigma}_q^2}{m} \right)$$

*is an unbiased estimator of $\ell$, where $\hat{\sigma}_p^2$ and $\hat{\sigma}_q^2$ are the sample variances of $f(x)$ under $p$ and $q$.*

*Proof.* Let $\mu_p = \mathbb{E}_{x\sim p}[f(x)]$, $\mu_q = \mathbb{E}_{x\sim q}[f(x)]$, $\sigma_p^2 = \text{Var}_{x\sim p}[f(x)]$, and $\sigma_q^2 = \text{Var}_{x\sim q}[f(x)]$. First, note that $\ell = (\mu_p - \mu_q - c)^2$. Let $\bar{X}_p = \frac{1}{n}\sum_{i=1}^{n} f(x_i^p)$ and $\bar{X}_q = \frac{1}{m}\sum_{j=1}^{m} f(x_j^q)$. We know that $\mathbb{E}[\bar{X}_p] = \mu_p$, $\mathbb{E}[\bar{X}_q] = \mu_q$, $\text{Var}[\bar{X}_p] = \frac{\sigma_p^2}{n}$, and $\text{Var}[\bar{X}_q] = \frac{\sigma_q^2}{m}$. Now, let's consider the expectation of $\hat{\ell}$:

$$\mathbb{E}[\hat{\ell}] = \mathbb{E}\left[ (\bar{X}_p - \bar{X}_q - c)^2 - \left( \frac{\hat{\sigma}_p^2}{n} + \frac{\hat{\sigma}_q^2}{m} \right) \right]$$

$$= \mathbb{E}\left[ (\bar{X}_p - \bar{X}_q - c)^2 \right] - \mathbb{E}\left[ \frac{\hat{\sigma}_p^2}{n} + \frac{\hat{\sigma}_q^2}{m} \right]$$

For the first term:

$$\mathbb{E}\left[ (\bar{X}_p - \bar{X}_q - c)^2 \right] = \text{Var}[\bar{X}_p - \bar{X}_q] + (\mathbb{E}[\bar{X}_p - \bar{X}_q - c])^2$$

$$= \frac{\sigma_p^2}{n} + \frac{\sigma_q^2}{m} + (\mu_p - \mu_q - c)^2$$

For the second term:

$$\mathbb{E}\left[ \frac{\hat{\sigma}_p^2}{n} + \frac{\hat{\sigma}_q^2}{m} \right] = \frac{\mathbb{E}[\hat{\sigma}_p^2]}{n} + \frac{\mathbb{E}[\hat{\sigma}_q^2]}{m}$$

$$= \frac{\sigma_p^2}{n} + \frac{\sigma_q^2}{m}$$

The last equality holds because the sample variance is an unbiased estimator of the population variance. Combining these results:

$$\mathbb{E}[\hat{\ell}] = \left( \frac{\sigma_p^2}{n} + \frac{\sigma_q^2}{m} + (\mu_p - \mu_q - c)^2 \right) - \left( \frac{\sigma_p^2}{n} + \frac{\sigma_q^2}{m} \right)$$

$$= (\mu_p - \mu_q - c)^2$$

$$= \ell$$

Therefore, $\hat{\ell}$ is an unbiased estimator of $\ell$. $\qquad\square$

**Proposition 2.** *Let $\mu_p = \mathbb{E}_{x \sim p}[f(x)]$, $\mu_q = \mathbb{E}_{x \sim q}[f(x)]$, $\sigma_p^2 = Var_{x \sim p}[f(x)]$, $\sigma_q^2 = Var_{x \sim q}[f(x)]$ and $n$ and $m$ be the number of independent samples from distributions $p$ and $q$, respectively. Then, the variance of the mini-batch estimator $\hat{\ell}$ is given by*

$$Var(\hat{\ell}) = \mathcal{O}\left( \left( \frac{\sigma_p^2}{n} + \frac{\sigma_q^2}{m} \right) \cdot \left( \frac{\sigma_p^2}{n} + \frac{\sigma_q^2}{m} + (\mu_p - \mu_q - c)^2 \right) \right).$$

*Proof.* Let $Y = \bar{X}_p - \bar{X}_q - c$, where $\bar{X}_p = \frac{1}{n} \sum_{i=1}^{n} f(x_i^p)$ and $\bar{X}_q = \frac{1}{m} \sum_{j=1}^{m} f(x_j^q)$. We have

$$\mathbb{E}[Y] = \mu_p - \mu_q - c, \quad \text{Var}(Y) = \frac{\sigma_p^2}{n} + \frac{\sigma_q^2}{m}.$$

To derive $\text{Var}(Y^2)$, we use the fact that

$$\text{Var}(Y^2) = \mathbb{E}[Y^4] - (\mathbb{E}[Y^2])^2.$$

First, we compute $\mathbb{E}[Y^2]$

$$\mathbb{E}[Y^2] = \text{Var}(Y) + (\mathbb{E}[Y])^2 = \frac{\sigma_p^2}{n} + \frac{\sigma_q^2}{m} + (\mu_p - \mu_q - c)^2.$$

Next, we compute $\mathbb{E}[Y^4]$

$$\mathbb{E}[Y^4] = 3 \left( \frac{\sigma_p^2}{n} + \frac{\sigma_q^2}{m} \right)^2 + 6 \left( \frac{\sigma_p^2}{n} + \frac{\sigma_q^2}{m} \right) (\mu_p - \mu_q - c)^2 + (\mu_p - \mu_q - c)^4 + \mathcal{O}(n^{-2}) + \mathcal{O}(m^{-2}).$$

Since $\text{Var}(Y^2) = \mathbb{E}[Y^4] - (\mathbb{E}[Y^2])^2$, we substitute the expressions

$$\text{Var}(Y^2) = 3 \left( \frac{\sigma_p^2}{n} + \frac{\sigma_q^2}{m} \right)^2 + 6 \left( \frac{\sigma_p^2}{n} + \frac{\sigma_q^2}{m} \right) (\mu_p - \mu_q - c)^2 + (\mu_p - \mu_q - c)^4$$

$$- \left( \frac{\sigma_p^2}{n} + \frac{\sigma_q^2}{m} + (\mu_p - \mu_q - c)^2 \right)^2 + \mathcal{O}(n^{-2}) + \mathcal{O}(m^{-2}).$$

Simplify the terms

$$\text{Var}(Y^2) = 2 \left( \frac{\sigma_p^2}{n} + \frac{\sigma_q^2}{m} \right)^2 + 4 \left( \frac{\sigma_p^2}{n} + \frac{\sigma_q^2}{m} \right) (\mu_p - \mu_q - c)^2 + \mathcal{O}(n^{-2}) + \mathcal{O}(m^{-2}).$$

Recall

$$\hat{\ell} = (\bar{X}_p - \bar{X}_q - c)^2 - \left( \frac{\hat{\sigma}_p^2}{n} + \frac{\hat{\sigma}_q^2}{m} \right).$$

The total variance is

$$\text{Var}(\hat{\ell}) = \text{Var}((\bar{X}_p - \bar{X}_q - c)^2) + \text{Var}\left( \frac{\hat{\sigma}_p^2}{n} + \frac{\hat{\sigma}_q^2}{m} \right).$$

Combining the results

$$\text{Var}\left(\frac{\hat{\sigma}_p^2}{n} + \frac{\hat{\sigma}_q^2}{m}\right) = \frac{2\sigma_p^4}{n(n-1)} + \frac{2\sigma_q^4}{m(m-1)}.$$

Therefore

$$\text{Var}(\hat{\ell}) = 2\left(\frac{\sigma_p^2}{n} + \frac{\sigma_q^2}{m}\right)^2 + 4\left(\frac{\sigma_p^2}{n} + \frac{\sigma_q^2}{m}\right)(\mu_p - \mu_q - c)^2 + \frac{2\sigma_p^4}{n(n-1)} + \frac{2\sigma_q^4}{m(m-1)} + \mathcal{O}(n^{-2}) + \mathcal{O}(m^{-2}).$$

By rearranging the terms, we get the final result. □

**Remark 1.** *For two independent and bounded random variables $X$ and $Y$, if $\mathbb{E}[X] - \mathbb{E}[Y] > 0$, then the probability $p(\sum_{i=1}^{k} X_i > \sum_{i=1}^{k} Y_i)$ will (approximately) increase as the sample size $k$ increases. Therefore, the multi-sample pairwise comparison is (approximately) more likely to be correct than the single-sample pairwise comparison. In the asymptotic setting ($k \uparrow \infty$), the probability will converge to 1 as $\mathbb{E}[X] > \mathbb{E}[Y]$.*

*Proof.* Since $\mathbb{E}[X] - \mathbb{E}[Y] > c$ for some positive constant $c$, we want to prove a lower bound for the probability $p\left(\sum_{i=1}^{k} X_i > \sum_{i=1}^{k} Y_i\right)$ as the sample size $k$ increases. Consider the sums $S_X = \sum_{i=1}^{k} X_i$ and $S_Y = \sum_{i=1}^{k} Y_i$. Let's denote $Z_i = X_i - Y_i$. Then we are interested in $p(S_X > S_Y)$, which is the same as $p\left(\sum_{i=1}^{k} Z_i > 0\right)$. By the linearity of expectation and given $\mathbb{E}[X_i] - \mathbb{E}[Y_i] > c$, we have,

$$\mathbb{E}[Z_i] = \mathbb{E}[X_i] - \mathbb{E}[Y_i] > c$$

The expectation of the sum of differences is

$$\mathbb{E}\left[\sum_{i=1}^{k} Z_i\right] = k\mathbb{E}[Z_i] > kc$$

By Hoeffding's inequality on $Z_i$ (assuming $Z_i \in [a, b]$, since $X$ and $Y$ are bounded), the probability that the sum deviates from its expectation by a certain amount is bounded by

$$p\left(\sum_{i=1}^{k} Z_i - \mathbb{E}\left[\sum_{i=1}^{k} Z_i\right] \leq -t\right) \leq \exp\left(\frac{-2t^2}{k(b-a)^2}\right)$$

Since we are interested in the probability $p\left(\sum_{i=1}^{k} Z_i > 0\right)$, we set $t = k\mathbb{E}[Z_i] = k\delta$, where $\delta = \mathbb{E}[Z_i]$. Let $\delta = \mathbb{E}[X] - \mathbb{E}[Y]$. Given $\delta > c$, we can apply Hoeffding's inequality to bound $p\left(\sum_{i=1}^{k} Z_i \leq 0\right)$:

$$p\left(\sum_{i=1}^{k} Z_i \leq 0\right) = p\left(\sum_{i=1}^{k} Z_i - k\delta \leq -k\delta\right)$$

Applying Hoeffding's inequality,

$$p\left(\sum_{i=1}^{k} Z_i - k\delta \leq -k\delta\right) \leq \exp\left(\frac{-2(k\delta)^2}{k(b-a)^2}\right) = \exp\left(\frac{-2k\delta^2}{(b-a)^2}\right)$$

Since $p\left(\sum_{i=1}^{k} Z_i > 0\right) = 1 - p\left(\sum_{i=1}^{k} Z_i \leq 0\right)$, we get,

$$p\left(\sum_{i=1}^{k} Z_i > 0\right) \geq 1 - \exp\left(\frac{-2k\delta^2}{(b-a)^2}\right)$$

Then the bound can be written as

$$p\left(\sum_{i=1}^{k} Z_i > 0\right) \geq 1 - \exp\left(\frac{-2k(\mathbb{E}[X] - \mathbb{E}[Y])^2}{(b-a)^2}\right)$$

For any $k$, the probability $p\left(\sum_{i=1}^{k} X_i > \sum_{i=1}^{k} Y_i\right)$ is at least,

$$p\left(\sum_{i=1}^{k} X_i > \sum_{i=1}^{k} Y_i\right) \geq 1 - \exp\left(\frac{-2k(\mathbb{E}[X] - \mathbb{E}[Y])^2}{(b-a)^2}\right)$$

Therefore, we can loosely speaking, the probability (lower bound) will increase as $k$ increases. The argument will surely hold in the asymptotic setting, i.e. $k \to +\infty$. $\qquad\square$

