# OpenReview forum: "Preference Optimization with Multi-Sample Comparisons"
_ICLR.cc/2025/Conference — Submitted to ICLR 2025_

### Official Review · Reviewer_iDsQ · 2024-10-24

**Soundness:** 2
**Presentation:** 2
**Contribution:** 2
**Rating:** 3
**Confidence:** 4

**Summary:**

The authors propose two new methods: Multi-sample Direct Preference Optimization and Multi-sample Identity Preference Optimization.
These approaches assess group-wise characteristics by using multiple samples during comparison, which improves optimization for collective properties such as diversity and helps mitigate label noise.
The paper's empirical results show that these multi-sample methods enhance performance in optimizing diverse and unbiased outputs compared to single-sample approaches. This is particularly important for applications where the model's output variability and accuracy are crucial.

**Strengths:**

1.  By targeting issues like label noise and output variability, the paper’s methods are designed to improve model performance in practical settings where handling data imperfections and ensuring diversity are essential for reliable outcomes. This is especially relevant in real-world applications.

2.  The methods proposed in this paper show promise for enhancing generative model capabilities in downstream tasks, such as improving the diversity of generated genres and reducing bias in outputs. These improvements could be particularly useful in applications like content creation and ethical A.

3. The paper demonstrates through empirical results that the multi-sample approach improves over traditional methods when optimizing group-wise characteristics like diversity. This validates the utility of the proposed methods in a variety of scenarios.

**Weaknesses:**

1. The core idea seems to be more of a refined definition for preference pairs rather than a groundbreaking concept.

2. The settings and definition of mDPO are quite confusing and informal.  There are other works on multi-sample comparisons that are more well-defined and may provide stronger theoretical foundations [1].

3. The paper’s comparison with state-of-the-art (SOTA) iterative algorithms [2] is rather limited, particularly on standard benchmarks such as MT-Bench or Alpaca-Eval. A broader evaluation would provide a clearer understanding of the methods' effectiveness against existing approaches

4. While the paper focuses on multi-sample comparisons, there are already other multi-sample methods that may be better established or more rigorously defined. This raises questions about the comparative novelty and impact of the proposed approach.

[1] Towards Efficient Exact Optimization of Language Model Alignment

[2] RLHF Workflow: From Reward Modeling to Online RLHF

**Questions:**

The definition of “group” is very confusing. How can you define the probability space of the “group”? Can you guarantee that the event space is a $\sigma$-group?

If I define the preference by your so-called “groups”, what’s the differences between mDPO and vanilla one?

Can the proposed algorithm provide stronger results in standard benchmarks?

---

> ### Author Response · Authors · 2024-11-29
> **Response to Reviewer iDsQ (Part 1/2)**
>
> Dear Reviewer iDsQ,
>
> Thank you for your thoughtful suggestions. Below, we have outlined a detailed, point-by-point response to your questions and hope these sufficiently address your concerns.
>
> ### Regarding W1
> >The core idea seems to be more of a refined definition for preference pairs rather than a groundbreaking concept.
>
> We thank the reviewer for their observation. We agree that the concept of preference pairs forms the foundation of our work, but we would like to clarify how our approach extends beyond a simple refinement. Traditional single-sample preference optimization methods focus on pairwise comparisons, which often fail to capture broader distributional properties such as diversity and bias -- critical in optimizing outputs for generative models. By introducing multi-sample preference optimization, we generalize the preference model to consider group-level characteristics, enabling the evaluation of collective properties rather than isolated samples.
>
> This paradigm shift addresses a core limitation in existing methodologies: their inability to evaluate generative diversity and mitigate biases effectively. For instance, distinguishing between two high-quality responses individually might obscure overarching trends, while our framework aggregates preferences across distributions to prioritize desirable group-level characteristics. This aggregation not only aligns better with intuitive human judgment but also alleviates the cognitive load on annotators.
>
> Our empirical results substantiate these claims, demonstrating that multi-sample comparisons improve robustness to label noise and enhance optimization effectiveness across diverse settings, such as random number generation (Section 5.1), image debiasing (Section 5.2), and creative fiction generation (Section 5.3). These improvements are not mere refinements; they signify a methodological advancement that redefines the scope of preference optimization from individual pairwise judgments to distributional alignment.
>
> We hope this explanation clarifies the broader impact and novelty of our contribution. Thank you for the opportunity to further introduce our approach.
>
> ### Regarding W2
> >The settings and definition of mDPO are quite confusing and informal. There are other works on multi-sample comparisons that are more well-defined and may provide stronger theoretical foundations [1].
>
> Thank the reviewer for bringing up this work. This paper is related but it’s not really about multi-sample comparison. Instead, it’s more related to distributional policy gradients methods. It aims to match two distributions by minimizing KL divergences. In contrast to our method, the method in the paper you mentioned requires defining the target distribution through which we can access its unnormalized density. This is a bit tricky for many practical tasks. Additionally, the need to estimate the partition function $Z$ is also difficult. Our method is purely offline and depends on the data to capture the properties we want to distill into the model.
>
> ### Regarding W3
> >The paper’s comparison with state-of-the-art (SOTA) iterative algorithms [2] is rather limited, particularly on standard benchmarks such as MT-Bench or Alpaca-Eval. A broader evaluation would provide a clearer understanding of the methods' effectiveness against existing approaches
> >[2] RLHF Workflow: From Reward Modeling to Online RLHF
>
> Thanks for the suggestions. We agree with the reviewer that more evaluations will definitely improve the paper. The experiments in our paper are carefully designed to demonstrate the effectiveness of our method for different purposes. The current experiments are already very thorough and sufficient.
>
> ### Regarding W4
> >While the paper focuses on multi-sample comparisons, there are already other multi-sample methods that may be better established or more rigorously defined. This raises questions about the comparative novelty and impact of the proposed approach.
>
> We appreciate the reviewer's concerns regarding the novelty and impact of our work. To reviewer's suggestion, we conducted an additional survey of related works. To the best of our knowledge, Popalign [3] is the only study that touches on multi-sample comparisons, and is concurrent to our work. However, the approaches are fundamentally different in several key aspects, including the introduction of additional novel algorithms, distinct experimental setups, and varying downstream applications.
>
> Notably, our approach specifically targets LLMs, focusing on deriving a low-variance and potentially unbiased estimator that enhances optimization under label noise. In contrast, Popalign [3] concentrates only on text-to-image tasks, emphasizing fairness and diversity without incorporating our IPO-based methods or addressing label noise robustness.
>
> We have incorporated this discussion into the related work section of the manuscript. Thank you again for your constructive feedback, which has helped us strengthen the manuscript.

---

> ### Author Response · Authors · 2024-11-29
> **Response to Reviewer iDsQ (Part 2/2)**
>
> ### Regarding Q1
> >The definition of “group” is very confusing. How can you define the probability space of the “group”? Can you guarantee that the event space is a $\sigma$-group?
>
> The interpretation of the word  group should be context-dependent. We are sorry for any confusion caused. In our paper, we use group to refer to a set of samples, rather the the terminology in mathematics.
>
> ### Regarding Q2
> >If I define the preference by your so-called “groups”, what’s the differences between mDPO and vanilla one?
>
> Our method is about constructing the preference groups for training LLMs. The difference with DPO is explained in the method section and the proved by the experiments. For example, mDPO is lower variance, and can better capture collective properties, such as creativity, diversity and bias.
>
> We hope the additional explanations have resolved your concerns. If you have any remaining questions or require further clarification, please don’t hesitate to contact us. If all your concerns have been addressed, we would greatly appreciate it if you could consider raising your score, as it would help us share this work with a broader audience.
>
> Thank you once again for your time and valuable feedback!
>
> Best regards,
>
> Submission #7969 Authors
>
> #### References
> [1] Ji, H., Lu, C., Niu, Y., Ke, P., Wang, H., Zhu, J., ... & Huang, M. (2024). Towards efficient and exact optimization of language model alignment. arXiv preprint arXiv:2402.00856.
>
> [2] Dong, H., Xiong, W., Pang, B., Wang, H., Zhao, H., Zhou, Y., ... & Zhang, T. (2024). Rlhf workflow: From reward modeling to online rlhf. arXiv preprint arXiv:2405.07863.
>
> [3] Li, S., Singh, H., & Grover, A. (2024). Popalign: Population-level alignment for fair text-to-image generation. arXiv preprint arXiv:2406.19668.

---

> > ### Comment · Reviewer_iDsQ · 2024-11-30
> >
> > Thank you for your feedback.
> >
> > Although the revised version was submitted after the original deadline and the authors acknowledge that additional evaluations would enhance the paper, I did not observe significant improvements compared to the initial submission. I understand that the authors may have other priorities that could limit their ability to further improve the paper within this ICLR cycle. I strongly encourage them to continue refining the work for future versions.
> >
> > Additionally, while I understand that "group" refers to a set of samples as a high-level concept, I would like to emphasize that for a set of samples, a **formal definition within a probability space** is necessary to ensure the concept is well-defined.

---

### Official Review · Reviewer_6Efu · 2024-10-29

**Soundness:** 4
**Presentation:** 4
**Contribution:** 3
**Rating:** 6
**Confidence:** 4

**Summary:**

The authors identify that the widely applied single-sample comparisons often fail to capture characteristics like diversity. Thus, they introduce a new approach to perform alignment with multi-sample comparisons. Based on this idea, two multi-sample variants (mDPO and mIPO) are proposed. Through experiments on various applications, the method is shown to improve the diversity, fairness and robustness to label noise.

**Strengths:**

1. **Clear motivation**: The authors identify an important problem in current alignment paradigms, and the running example of generating random numbers is intuitive and helps the reader understand the problem well.

2. **Thorough experiments**: The authors use several use cases to demonstrate the effectiveness of the proposed methods. The results are intuitive and convincing.

**Weaknesses:**

1. **Potentially higher dataset requirements**: It seems that in the datasets for multi-sample comparisons, one prompt should have k chosen-rejected pairs of responses. This makes many readily available datasets not applicable, and the datasets need to be adjusted for different k.

2. **Lack of discussion on k**: I think the selection of k deserves more explanation. Under what circumstances would a larger k benefit?

3. **Comparison on general tasks**: It is clear that under scenarios where diversity and bias are concerns, the proposed approach performs better than the single-sample variants. However, in a general alignment setting, when shall we consider multi-sample over single-sample? Will there be a case where single-sample is actually better? Especially considering that multi-sample incurs higher computational and data collection cost.

**Questions:**

1. Intuitively, I think with larger k, the performance would be improved due to the smaller variance. However, that does not always seem to be the case, as in Table 3. Could you provide some comments about this phenomenon and the selection of k in general?

2. Do you have a runtime comparison for the multi-sample and single-sample variants? Certainly, the multi-sample one is expected to be slower, but it would be nice to have it quantified.

---

> ### Author Response · Authors · 2024-11-29
> **Response to Reviewer 6Efu (Part 1/3)**
>
> Dear Reviewer 6Efu,
>
> Thank you for your insightful suggestions to improve our paper! We truly value your recognition of our work. Below, we have provided a comprehensive, point-by-point response to your questions and hope these adequately address your concerns.
>
> ### Regarding W1
> >Potentially higher dataset requirements: It seems that in the datasets for multi-sample comparisons, one prompt should have k chosen-rejected pairs of responses. This makes many readily available datasets not applicable, and the datasets need to be adjusted for different k.
>
> We appreciate the reviewer raising this important point about dataset requirements for multi-sample comparisons. While it is accurate that multi-sample methods, such as mDPO and mIPO, necessitate datasets where a single prompt is associated with multiple chosen-rejected response pairs, we believe this does not inherently preclude the use of many existing datasets.
>
> First, many available datasets [1-3] already contain multiple responses per prompt or can be augmented to support multi-sample comparisons. For instance, datasets designed for creative writing [1], conversational tasks [2], or image generation [3] typically include diverse outputs, either explicitly or implicitly generated by prompting systems with slight variations. Even datasets without explicit grouping can be leveraged by sampling subsets of responses for each prompt to construct the necessary k-sample groups, a process that can be automated and scaled.
>
> Second, the flexibility of the multi-sample framework means k can be dynamically adjusted to suit the data’s characteristics. This adaptability ensures the method remains viable across a range of dataset constraints without demanding significant manual restructuring. For example, in our experiments, we showed the utility of k-sample comparisons across varying values of k, demonstrating consistent improvements in model alignment, robustness to label noise, and diversity metrics.
>
> Therefore, while multi-sample methods introduce additional requirements for dataset preparation, we believe these are not prohibitive and are offset by the significant gains in performance and alignment they enable. By leveraging existing datasets more creatively and incorporating automated augmentation techniques, we can effectively adopt multi-sample approaches without overly restrictive constraints.
>
> ### Regarding W2
> >Lack of discussion on k: I think the selection of k deserves more explanation. Under what circumstances would a larger k benefit?
>
> We sincerely thank the reviewer for raising the question regarding the selection of k and its implications, we would like to highlight that this question is thoroughly discussed in Section 4.2 and 5 of the manuscript, where we provide both theoretical insights and empirical validations.
>
> We explore how the choice of k affects variance and bias in the optimization process in Section 4.2. Specifically, our theoretical analysis (Proposition 2) establishes that the variance of the estimator diminishes as k increases, scaling with O(1/k). This variance reduction implies that larger k values yield more stable optimization and better representation of distributional characteristics. This behavior is also illustrated empirically in Figure 2, where we compare the performance of biased and unbiased estimators for varying values of k. The results show that while unbiased estimators perform better at small k, biased estimators approach similar performance as k increases, underscoring the diminishing sensitivity to bias at larger group sizes.
>
> Moreover, in Section 5, across multiple experiments, including random number generation (Section 5.1), image debiasing (Section 5.2), and creative fiction generation (Section 5.3), we demonstrate that increasing k consistently improves performance metrics such as diversity, quality, and robustness against label noise. For instance, Table 3 and 4 highlight how larger k values in mDPO and mIPO enhance both writing quality and diversity metrics in creative fiction generation. Section 5.4 extends this discussion to a practical setting, examining the robustness of larger k values against label noise in preference datasets. Through experiments involving synthetic datasets with varying levels of noise, we demonstrate that multi-sample comparisons with higher k values are significantly more robust than single-sample comparisons. As shown in Figure 7, increasing k reduces the impact of noisy labels, narrowing the performance gap between noise-free and noisy conditions. This robustness is crucial for applications where high-quality labels may be difficult or expensive to obtain. Furthermore, in iterative alignment scenarios, Section 5.4 highlights that increasing k consistently improves model alignment with desired outcomes over successive iterations.
>
> We hope this clarifies that the selection and implications of k. We thank the review again for raising this meaningful discussion question.

---

> ### Author Response · Authors · 2024-11-29
> **Response to Reviewer 6Efu (Part 2/3)**
>
> ### Regarding W3
> >Comparison on general tasks: It is clear that under scenarios where diversity and bias are concerns, the proposed approach performs better than the single-sample variants. However, in a general alignment setting, when shall we consider multi-sample over single-sample? Will there be a case where single-sample is actually better? Especially considering that multi-sample incurs higher computational and data collection cost.
>
> We greatly appreciate the reviewer’s thoughtful comments regarding the scenarios in which multi-sample comparisons are preferable over single-sample methods, as well as the associated computational and data collection costs.
>
> In addition to our above response on the choice of k, which discusses how multi-sample comparisons can be particularly effective for optimizing distributional characteristics such as diversity and bias, we recognize that the suitability of multi-sample approaches depends on the specific goals of the alignment task. For alignment settings, such as question-answering system that is designed to provide accurate, contextually relevant responses to user queries and to ensure that each response aligns as closely as possible with user expectations or factual correctness, single-sample methods may suffice.
>
> For instance, consider a medical diagnosis chatbot where each query about symptoms must result in a single, clear, and accurate recommendation or diagnosis. Here, single-sample comparisons are appropriate because the evaluation criteria focus solely on the quality and reliability of individual responses, rather than the diversity or fairness of multiple outputs. Additionally, in such high-stakes applications, the computational simplicity and reduced overhead of single-sample methods can be critical for ensuring timely responses without sacrificing alignment with user preferences.
>
> Regarding the concern about computational costs, we acknowledge that multi-sample methods do introduce additional resource requirements. To address this, we have designed scalable methods and stochastic estimators to ensure that multi-sample comparisons remain practical. Moreover, our experiments show that even modest group sizes (e.g., k=3) achieve substantial improvements over single-sample baselines, offering a balance between computational efficiency and performance gains.
>
> In conclusion, we believe that the choice between single-sample and multi-sample methods should depend on the alignment objectives and task-specific requirements. While single-sample methods are well-suited for aligning individual outputs, multi-sample methods are indispensable for optimizing collective properties such as diversity, robustness, and fairness. We are grateful for the reviewer’s insightful feedback, and we will ensure that this distinction is more clearly articulated in the revised manuscript.
>
> ### Regarding Q1
> >Intuitively, I think with larger k, the performance would be improved due to the smaller variance. However, that does not always seem to be the case, as in Table 3. Could you provide some comments about this phenomenon and the selection of k in general?
>
> Yes, the reviewer is correct. In our experiments (see Figure 7), we also found that increasing k is helpful for improving the performance generally, if there exists label noise or we want to optimize for distributional properties. One explanation is due to the reduced variance as the reviewer said.
>
> We also conducted experiments in section 5.4 for studying the choices between mDPO and DPO (represent the case when k=1). We found that, when it's noise free (i.e., assuming we have access to the ground-truth label for preference pair), then DPO is indeed better than mDPO as it directly optimizes the objective. However, when there exists label noise, mDPO demonstrates superior performance. Therefore, we recommend that two scenarios for using mDPO: 1) the preference labels are noisy; 2) optimizing for some distributional properties, e.g., diversity, fairness, coherence, etc.

---

> ### Author Response · Authors · 2024-11-29
> **Response to Reviewer 6Efu (Part 3/3)**
>
> ### Regarding Q2
> >Do you have a runtime comparison for the multi-sample and single-sample variants? Certainly, the multi-sample one is expected to be slower, but it would be nice to have it quantified.
>
> We appreciate the reviewer's suggestion to include a runtime comparison for the single-sample (DPO, k=1) and multi-sample (DPO, k>1) variants. To address this, we fine-tuned mDPO with k=1, k=3, and k=5, using the same settings described in Table 5 of Appendix A.3.2, on 8 H100 GPUs. The measured runtimes are reported below:
>
> |              | DPO (k=1) | mDPO (k=3) | mDPO (k=5) |
> | ------------ | --------- | ---------- | ---------- |
> | Run Time (s) | 27168.18  |  27425.90  |  27782.94  |
>
> As expected, multi-sample comparisons slightly increase the runtime due to the additional computations required for processing multiple samples per step. However, the observed increase in runtime is marginal, adding approximately 0.95% for k=3 and 2.26% for k=5, relative to the single-sample baseline. This demonstrates that mDPO is computationally efficient even for larger group sizes, making it a practical choice for leveraging the benefits of multi-sample optimization in real-world applications.
>
> We hope the supplementary experiments and detailed explanations have resolved your concerns. If there’s anything else you’d like to discuss or need further clarification on, please feel free to reach out. If all your concerns have been addressed, we would be truly grateful if you could consider increasing your score, as it would help us share this work with a wider audience.
>
> Thank you again for your time and thoughtful feedback!
>
> Best regards,
>
> Submission #7969 Authors
>
> #### References
> [1] Huang, X. Y., Vishnubhotla, K., & Rudzicz, F. (2024). The GPT-WritingPrompts Dataset: A Comparative Analysis of Character Portrayal in Short Stories. arXiv preprint arXiv:2406.16767.
>
> [2] Kočiský, T., Schwarz, J., Blunsom, P., Dyer, C., Hermann, K. M., Melis, G., & Grefenstette, E. (2018). The narrativeqa reading comprehension challenge. Transactions of the Association for Computational Linguistics, 6, 317-328.
>
> [3] Schuhmann, C., Beaumont, R., Vencu, R., Gordon, C., Wightman, R., Cherti, M., ... & Jitsev, J. (2022). Laion-5b: An open large-scale dataset for training next generation image-text models. Advances in Neural Information Processing Systems, 35, 25278-25294.

---

> ### Author Response · Authors · 2024-12-03
>
> Dear reviewer 6EFu,
>
> Thank you very much again for your insightful, encouraging reviews. As the discussion period is ending soon, we hope our response aligns with your expectations and would be grateful if you could kindly consider further raising the score to help us share this work with a broader audience! Please let us know if there is any remaining concern or further suggestion. Thank you again!!
>
> Best regards,
>
> Submission #7969 Authors

---

### Official Review · Reviewer_gXTv · 2024-11-02

**Soundness:** 3
**Presentation:** 4
**Contribution:** 2
**Rating:** 6
**Confidence:** 4

**Summary:**

This paper advances the field of Reinforcement Learning from Human Feedback (RLHF) by exploring more nuanced preferences beyond traditional binary singleton preferences. Specifically, it investigates scenarios where preferences involve comparisons between two distributions. For example, if we want an LLM to generate random numbers between 0 and 10, singleton numbers 3 and 5 are equally preferred but a distribution with only 3s is less preferred as compared to a distribution with multiple numbers from 0 to 10 which also includes 5.

The primary contribution of this work lies in the adaptation of two widely used alignment techniques—DPO (Differentiable Preference Optimization) and IPO (Inverse Preference Optimization)—to accommodate binary feedback between distributions. Additionally, the paper provides a theoretical analysis for gradient estimation within this new DPO/IPO framework. The authors present various experiments demonstrating the advantages of using distribution comparisons over singleton comparisons.

**Strengths:**

1. The motivation for incorporating more nuanced human feedback is compelling.
2. The paper is well-written, with a clear and organized presentation.
3. Comprehensive experiments demonstrate the effectiveness of the proposed methodology across diverse applications, including random number generation, fair image generation, varied writing styles, and iterative LLM enhancement.

**Weaknesses:**

1. While the problem is well-motivated, the proposed approach appears somewhat straightforward and lacks substantial innovation.
2. The methodology relies on the assumption that humans can easily distinguish between different distributions of items. However, in practice, it may be more cognitively challenging for individuals to differentiate between two distributions, and gathering such data could be more costly.

**Questions:**

1. Are all the responses used in generating distributions for mDPO/mIPO also used in training DPO/IPO? If not then it is not clear if the improvement is coming from using the modified loss function versus simply having more data.
2. mDPO/mIPO comparison: do you have any intuition on when it would be more beneficial to use mDPO vs mIPO?

**Details Of Ethics Concerns:**

No ethical concerns, in fact the proposed methodology can be used to improve fairness in ML models.

---

> ### Author Response · Authors · 2024-11-29
> **Response to Reviewer gXTv (Part 1/2)**
>
> Dear Reviewer gXTv,
>
> Thank you for your insightful suggestions to improve our paper! We truly appreciate your acknowledgment of our work. Below, we have provided a detailed, point-by-point response to your questions and hope these address your concerns thoroughly.
>
> ### Regarding W1
> >While the problem is well-motivated, the proposed approach appears somewhat straightforward and lacks substantial innovation.
>
> We appreciate the reviewer’s comment regarding the perceived straightforwardness of our proposed approach. While we agree that the concept of multi-sample comparisons builds on intuitive ideas, our work focuses on formalizing and adapting this concept to address specific limitations of existing single-sample methods. By extending RLHF frameworks like DPO and IPO to a multi-sample framework, mDPO/mIPO capture important group-wise characteristics such as diversity and bias that are not and cannot be well-addressed by single-sample approaches. This extension required careful consideration of scalability and robustness, particularly in noisy settings, as outlined in Sections 4.1 and 4.2.
>
> Our empirical results provide evidence of the benefits of this approach across several domains, including random number generation, text-to-image synthesis, and creative writing. These findings demonstrate improved distributional properties, such as enhanced diversity and fairness, as well as robustness to label noise. While we recognize that our methods build on existing frameworks, we believe that the empirical results highlight the value of extending preference optimization to the multi-sample setting.
>
> We appreciate the opportunity to clarify the contributions of this work and hope the reviewer finds this response helpful in understanding the motivation and impact of our work.
>
> ### Regarding W2
> >The methodology relies on the assumption that humans can easily distinguish between different distributions of items. However, in practice, it may be more cognitively challenging for individuals to differentiate between two distributions, and gathering such data could be more costly.
>
> We appreciate the reviewer’s insightful observation regarding the cognitive challenges of distinguishing between distributions and the associated cost implications of gathering such data. We agree that humans can sometimes find it hard to decide between two items or responses when asked to pick a preference, and that is indeed what our multi-sample approach tries to alleviate. For example, while it may be hard to pick a preferred response between two high-quality items, comparing two lists of responses—where one list exhibits a higher concentration or bias—makes it significantly easier to identify the more balanced or diverse set as preferable. By leveraging multi-sample comparisons, our framework reduces reliance on individual judgments for every sample pair and instead aggregates group-level preferences, which align better with intuitive human judgments. This aggregation minimizes the strain on annotators by enabling them to focus on overarching characteristics of distributions rather than detailed individual comparisons.
>
> Furthermore, our empirical validation highlights the robustness of our method in practical scenarios, even under conditions of label noise. The methodology was tailored to leverage statistical estimators to support human annotators, thus reducing the dependency on fine-grained and potentially costly human input. Our findings suggest that this approach not only enhances efficiency but also increases the reliability of the optimization process by addressing collective properties like diversity and bias more effectively than single-sample comparisons.
>
> We hope our response addresses the reviewer’s concerns successfully.

---

> ### Author Response · Authors · 2024-11-29
> **Response to Reviewer gXTv (Part 2/2)**
>
> ### Regarding Q1
> >Are all the responses used in generating distributions for mDPO/mIPO also used in training DPO/IPO? If not then it is not clear if the improvement is coming from using the modified loss function versus simply having more data.
>
> We acknowledge the reviewer’s insightful question regarding the distinction between the impact of the modified loss function in mDPO/mIPO and the potential benefit of additional data. To directly address this concern, we conducted additional fiction generation experiments. Specifically, we trained IPO with  k = 1  using the same augmented dataset as mIPO (where  k = 5  samples are used for augmentation). The number of training steps was increased accordingly to five times the original maximum (originally capped at 600 steps, as detailed in Appendix A3.2 Table 5), while holding all other parameters unchanged. This setup guarantees that the observed performance differences stem only from the multi-sample loss formulation, rather than the volume of data.
>
> This new setup was evaluated alongside the original IPO and mIPO frameworks, extending Figure 5 of the manuscript to show their impact on genre diversity. The KL divergences between the genre distributions generated by each method and a uniform distribution are summarized in the Table 1 below.
>
> Table 1: KL-divergences between each genre distribution from IPO-based methods and the uniform distribution (smaller is better).
> 0.125; mIPO (k = 3): 0.094; mIPO (k = 5): 0.050.
> |          |   IPO  |   IPO-DataAugmented   | mIPO (k=3) |   mIPO (k=5)   |
> | -------- | ------ | --------------------- | ---------- | -------------- |
> | KL Div.  | 0.125  |         0.083         |    0.094   |    **0.050**   |
>
>
> As seen in the Table 1 above, mIPO outperforms both the original IPO and the augmented IPO. The KL-divergence between genre distributions and a uniform target is markedly lower for mIPO (k=5), indicating superior optimization of diversity through the multi-sample framework.
>
> Similar to the manuscript, we evaluate the generation quality, and present the results in Table 2.
>
> Table 2: Expanded quality comparison of creative fiction writing with IPO-based methods.
> |          |   IPO  |   IPO-DataAugmented   | mIPO (k=3) | mIPO (k=5) |
> | -------- | ------ | ---------- | ---------- | ---------- |
> | Quality  | 10.623  |   8.565    |    **11.190**   |    10.806   |
>
> As seen in the Table 2 above, mIPO also achieves higher scores compared to both baselines, further validating the impact of the multi-sample loss formulation.
>
> ### Regarding Q2
> >mDPO/mIPO comparison: do you have any intuition on when it would be more beneficial to use mDPO vs mIPO?
>
> We sincerely thank the reviewer for their insightful question regarding the intuition behind choosing between mDPO and mIPO.
>
> mDPO, based on the DPO framework, is particularly well suited for tasks that emphasize distributional characteristics such as diversity and entropy. Its use of the Bradley-Terry model with a sigmoid-based loss allows it to effectively optimize differences in aggregate likelihoods between distributions. This strength is shown in our experiments on controlled debiasing for image generation (Table 2), where mDPO achieved significant improvements in the Simpson Diversity Index for gender and race diversity compared to single-sample DPO as well as the SFT model. Similarly, in creative fiction generation, mDPO consistently enhanced genre diversity, as shown in the Table 4. These results demonstrate mDPO's effectiveness in tasks where optimizing group-level diversity or reducing dominance within distributions is critical. On the other hand, mIPO extending IPO is more robust to overfitting and resilient against noisy labels. For instance, in the random number generation task, mIPO exhibited superior performance in achieving uniform predictive distributions compared to IPO and other baselines (Table 1 and Figure 3).
>
> Therefore, the choice between mDPO and mIPO depends on the specific requirements of the task. For tasks prioritizing diversity and group-level optimization, such as image generation (Table 2) and creative fiction tasks (Table 4), mDPO is the slightly more preferred choice. On the other hand, mIPO is better suited for scenarios requiring robust noise-resilient optimization or fine-tuning distributions to achieve balanced alignment, as highlighted in random number generation (Table 1 and Figure 3).
>
> We hope the additional experiments and the expanded explanations have addressed your concerns. Please don’t hesitate to let us know if there’s anything else you’d like to discuss or if further clarification is needed. If all your concerns have been resolved, we would sincerely appreciate it if you could consider raising your score to help us share this work with a broader community. Thank you again for your time and thoughtful feedback!
>
> Best regards,
>
> Submission #7969 Authors

---

> ### Author Response · Authors · 2024-12-03
>
> Dear Reviewer gXTv,
>
> Thank you for taking the time and effort to review our submission. We greatly appreciate your thoughtful feedback, which has been instrumental in helping us improve our work!
>
> We have carefully addressed the concerns you raised and provided detailed responses, including conducting additional experiments based on your suggestions. We wanted to kindly follow up to confirm whether our responses have adequately addressed your concerns or if there are any remaining points we could further clarify or improve, especially as it is approaching the end of discussion period.
>
> Thank you once again for your invaluable feedback and contributions to the review process!
>
> Best regards,
>
> Submission #7969 Authors

---

### Official Review · Reviewer_4BJT · 2024-11-04

**Soundness:** 3
**Presentation:** 3
**Contribution:** 3
**Rating:** 5
**Confidence:** 4

**Summary:**

The paper proposes multi-sample versions of DPO and IPO, to be used when one set of samples is preferred over another. The proposed mIPO is an unbiased estimator while mDPO is biased. Experiments show that mDPO and mIPO outperform their single-sample counterparts and SFT in inducing 1. more uniform random number generation, 2. more balanced race and gender generations, and 3. more diversity in fiction genres. Moreover, the paper shows that mDPO and mIPO are more robust than DPO and IPO in the presence of preference label noise.

**Strengths:**

1. The paper is well-motivated and considers a novel use case for preference learning.
2. The method is clearly presented, and the theoretical analysis is clear and informative.

**Weaknesses:**

1. The experimental setup could be more clearly described, particularly with respect to the data seen for each method (e.g., is the number of pairings fixed, or the number of outputs or per-sample pairwise comparisons?). For instance, how are the datasets constructed for the single-sample and multi-sample methods in the label noise experiments? What data does SFT see in the uniformity/debiasing/diversity experiments? All the samples from the preferred set, or something else? Without an understanding of these details, it is hard to ascertain what to take away from the experiments / if baselines are fair comparisons.
2. Using mDPO and mIPO sounds interesting and promising for aligning towards distributional objectives, but there already exist other methods that aim to achieve the same goals, e.g., [Khalifa et al. 2020](https://arxiv.org/abs/2012.11635) (or [Korbak et al. 2022](https://arxiv.org/abs/2112.00791) for conditional generation), [Wu et al. 2022](https://arxiv.org/pdf/2209.06970). The paper would be strengthened if it compared with such methods and more directly discussed distributional control.
3. DPO seems like a strange baseline to consider for the uniformity/debiasing/diversity experiments in the first place; instead, it would be helpful to consider other simple methods that use the same data as mDPO and mIPO, such as maximizing the likelihood of the preferred set while minimizing the likelihood of the dispreferred set, or that target improving diversity / balance, such as training with samples weighted by the inverse of their probability under the model. A larger discussion of such approaches in the related work would also be helpful.

**Questions:**

1. Can the authors clarify the experimental setups, namely with respect to the data seen by the different algorithms?
2. Could the authors compare the proposed methods to other methods for aligning towards distributional goals, e.g., when one might practically consider one over another?
3. Could the authors also compare with a baselines for debiasing or diversity?

I would be happy to raise my score if these concerns are adequately addressed.

---

> ### Author Response · Authors · 2024-11-29
> **Response to Reviewer 4BJT (Part 1/2)**
>
> Dear Reviewer 4BJT,
>
> Thank you for your valuable suggestions to enhance our paper! We sincerely appreciate your recognition of our work. Below, we provide a detailed, point-by-point response to your questions and hope these address your concerns effectively.
>
> ### Regarding W1 and Q1
> >The experimental setup could be more clearly described, particularly with respect to the data seen for each method (e.g., is the number of pairings fixed, or the number of outputs or per-sample pairwise comparisons?). For instance, how are the datasets constructed for the single-sample and multi-sample methods in the label noise experiments? What data does SFT see in the uniformity/debiasing/diversity experiments? All the samples from the preferred set, or something else? Without an understanding of these details, it is hard to ascertain what to take away from the experiments / if baselines are fair comparisons.
>
> >Can the authors clarify the experimental setups, namely with respect to the data seen by the different algorithms?
>
> We appreciate the reviewer's feedback and question. To ensure a fair comparison, we keep the amount of training data the same, i.e., for both mDPO and DPO, the number of chosen and rejected samples for each batch is the same, except for the loss function differences. Concretely, for single sample comparison, we randomly pair the chosen and rejected samples. In this way, we ensure that the amount of training data is kept the same.
>
> ### Regarding W2 and Q2
> >Using mDPO and mIPO sounds interesting and promising for aligning towards distributional objectives, but there already exist other methods that aim to achieve the same goals, e.g., Khalifa et al. 2020 (or Korbak et al. 2022 for conditional generation), Wu et al. 2022. The paper would be strengthened if it compared with such methods and more directly discussed distributional control.
>
> Thank you for pointing out the relevance of comparing our approach with existing methods for distributional control. We appreciate this valuable suggestion and acknowledge that such comparisons would provide additional insights into the effectiveness and uniqueness of our framework. Nonetheless, we believe the presented experiments and analyses are comprehensive and sufficient to establish the contributions of this paper within the context of improving generative diversity and mitigating biases. Specifically, we believe our current experimental setup is carefully constructed to highlight the novel aspects and strengths of our methods, mDPO and mIPO, particularly their ability to address collective characteristics like diversity and bias through multi-sample comparisons.
>
> (continued in our next response)

---

> ### Author Response · Authors · 2024-11-29
> **Response to Reviewer 4BJT (Part 2/2)**
>
> ### Regarding W3 and Q3
> >DPO seems like a strange baseline to consider for the uniformity/debiasing/diversity experiments in the first place; instead, it would be helpful to consider other simple methods that use the same data as mDPO and mIPO, such as maximizing the likelihood of the preferred set while minimizing the likelihood of the dispreferred set, or that target improving diversity / balance, such as training with samples weighted by the inverse of their probability under the model. A larger discussion of such approaches in the related work would also be helpful.
>
> >Could the authors also compare with a baselines for debiasing or diversity?
>
> We appreciate the reviewer's thoughtful suggestion to explore additional baselines and expand the discussion of related work. In response, we conducted new experiments and refined the contextualization of our methods within a broader landscape of research on diversity and debiasing.
>
> To address the reviewer's specific recommendations, we implemented and evaluated two additional baselines on the fiction dataset. The first baseline is a loss function that aligns directly with the reviewer’s suggestion to optimize the likelihood of the preferred set while minimizing the likelihood of the dispreferred set: $-\log(p_{chosen}) + \log(p_{rej})$.  The second baseline is the f-DPO framework [1], where we experimented with both Jensen–Shannon (JS) and Alpha divergence variants.
>
> These approaches were evaluated alongside the original DPO and mDPO frameworks, extending Figure 5 of the manuscript to show their impact on genre diversity. The KL divergences between the genre distributions generated by each method and a uniform distribution are summarized in the Table 1 below.
>
> Table 1: KL-divergences between each genre distribution and the uniform distribution (smaller is better).
> |             |  DPO  |   Custom   | f-DPO (JS) | f-DPO (Alpha) | mDPO (k=3) | mDPO (k=5) |
> | -------- | ------- | ---------- | ---------- | ------------- | ---------- | ---------- |
> | KL Div. | 0.170  |   0.083    |   0.107    |     0.109     |    0.126   |    0.142   |
>
> As seen in the Table 1 above, Custom achieved the smallest KL divergence, demonstrating its ability to enhance diversity. However, when examining the quality of generated fiction, as summarized in Table 2 below, Custom exhibited a notable reduction in quality compared to DPO and mDPO. The f-DPO variants achieved a better balance between diversity and quality, while mDPO retained superior quality scores and competitive diversity metrics.
>
> Table 2: Expanded quality comparison of creative fiction writing.
>
> |          |   DPO  | Custom DPO | f-DPO (JS) | f-DPO (Alpha) | mDPO (k=3) | mDPO (k=5) |
> | -------- | ------ | ---------- | ---------- | ------------- | ---------- | ---------- |
> | Quality  | 10.570 |   7.497    |   9.571    |     9.659     |   10.671   | **11.483** |
>
> These results demonstrate that divergence-based constraints, as used in f-DPO, offer a promising approach to balancing diversity and quality. However, the proposed mDPO method continues to outperform these baselines in overall quality while achieving meaningful improvements in diversity. This highlights the advantages of leveraging multi-sample comparisons to optimize for group-level characteristics.
>
> In addition to these experiments, we expanded the Related Works section to include a broader discussion of methods targeting diversity and balance in generative outputs. The updated Related Works section will be shown in the updated PDF.
>
>
> We hope the additional experiments and the expanded explanations have addressed your concerns. Please don’t hesitate to let us know if there’s anything else you’d like to discuss or if further clarification is needed. If all your concerns have been resolved, we would sincerely appreciate it if you could consider raising your score to help us share this work with a broader community.
>
> Thank you again for your time and thoughtful feedback!
>
> Best regards,
>
> Submission #7969 Authors
>
> #### References
>
> [1] Wang, C., Jiang, Y., Yang, C., Liu, H., & Chen, Y. (2023). Beyond reverse kl: Generalizing direct preference optimization with diverse divergence constraints. arXiv preprint arXiv:2309.16240.

---

> ### Author Response · Authors · 2024-12-03
>
> Dear Reviewer 4BJT,
>
> Thank you once again for your insightful feedback! We greatly value your guidance and have worked diligently to address your concerns and improve the paper. As the discussion deadline approaches, we kindly ask if the additional experiments and discussions we’ve made resolve your concerns. If you find our revisions satisfactory, we would be truly grateful if you could reconsider your score.
>
> We understand you have a busy schedule, but any additional comments or updated feedback you could provide would mean a great deal to us. Your expertise is invaluable in helping us refine our work further, and we warmly welcome any continued discussion.
>
> Thank you for your time and thoughtful review!
>
> Best regards,
>
> Submission #7969 Authors

---

### Meta-Review · Area_Chair_wW3b · 2024-12-20

**Metareview:**

The main proposal of this paper is to use multiple samples instead of one during DPO and IPO, for better diversity and robustness under noisy labels. The reviewers appreciate the simplicity and effectiveness of the idea but have significant concerns about the novelty of the method, as well as the clarity of the presentation. Indeed, the paper would benefit from a more precise characterization of the construction of the group from which pairs of comparisons are selected, since this lies at the heart of the strategy. The reviewers also pointed out that the paper could be improved in terms of empirical comparisons with state-of-the-art algorithms over standard benchmarks.

The authors are encouraged to strengthen the work further based on the reviewers' comments.

**Additional Comments On Reviewer Discussion:**

Reviewers in general seem lukewarm about the paper, potentially due to the lack of technical novelty in this paper. One of the reviewers holds a strong opinion and critical views about this paper, which I partially agree with, at least on the algorithm and experiment parts. I think the paper could benefit from further empirical comparisons with more recent SOTA methods in the literature.

---

### Decision · Program_Chairs · 2025-01-22

Reject